# Mechanosensitive channel engineering: A study on the mixing and matching of YnaI and MscS sensor paddles and pores

Vanessa J. Flegler[1,2], Akiko Rasmussen[1,2], Rainer Hedrich[3,4,5], Tim Rasmussen [1,2] & Bettina Böttcher [1,2] ✉

Osmotically varying environments are challenging for bacterial cells. Sudden drops in osmolytes cause an increased membrane tension and rupture the cells in the absence of protective mechanisms. One family of protective proteins are mechanosensitive channels of small conductance that open in response to membrane tension. Although these channels have a common architecture, they vary widely in the number of transmembrane helices, conductivity, and gating characteristics. Although there are various structures of channels in the open and closed state, the underlying common principles of the gating mechanism remain poorly understood. Here we show that YnaI opens by radial relocation of the transmembrane sensor paddles together with a shortening of the pore, which contrasts the prototypic smaller MscS. A chimera of both channels with the YnaI sensor paddles and the pore containing C-terminal part of MscS is functional and has the tension response of the paddle donor. Our research shows that elements with different structural opening mechanisms can be mixed and matched within one channel as long as they support the common area expansion on the periplasmic side.

Bacteria are exposed to many changes in their environment and must adapt quickly to survive. One of these challenges is the maintenance of osmotic homoeostasis. A hypoosmotic shock will rapidly drive water into the cell, eventually tensing the membrane up to the point of rupture. To prevent cell lysis, bacteria have mechanosensitive channels that are gated in response to the elevated membrane tension and release solutes, restoring the turgor within the cells[1–3]. Different types of mechanosensitive channels convey a graded response mechanism. One family is designated as mechanosensitive channels of small conductance because of their electrophysiological properties: They typically open at lower membrane tension levels than the mechanosensitive channel of large conductance, MscL, and release only small solutes with a smaller flux. One of the first discovered family

members is MscS[1], which is a homo-heptamer comprised of a cytosolic vestibule and a transmembrane part with a central pore surrounded by seven sensor paddles. In addition, *Escherichia coli* (*E. coli*) has five larger-sized paralogues with a similar overall architecture but more transmembrane (TM) helices and different gating characteristics[4–6].

One of the medium-sized paralogues is called YnaI, with 5 TM helices. YnaI opens at a larger membrane tension and with a smaller conductivity than all other MscS-like paralogues in *E. coli*[4,7,8]. Therefore, its importance for the protection against osmotic challenges is enigmatic as only a few solutes will leave the cell via YnaI in the presence of the other paralogues. Structural studies have shown that the main difference between YnaI and MscS are extended sensor paddles, which create a periplasmic indentation in YnaI, and much smaller

[1]Julius-Maximilians-Universität Würzburg, Biocenter and Rudolf-Virchow-Center, Josef-Schneider-Str. 2, Gebäude D15, Würzburg, Germany. [2]Julius-Maximilians-Universität Würzburg, Department of Biochemistry II, Am Hubland, Würzburg, Germany. [3]Faculty of Synthetic Biology, Shenzhen University of Advanced Technology, Shenzhen, China. [4]Institute of Emerging Agricultural Technology, Shenzhen University of Advanced Technology, Shenzhen, China. [5]State Key Laboratory of Quantitative Synthetic Biology, Shenzhen Institute of Synthetic Biology, Shenzhen Institutes of Advanced Technology, Chinese Academy of Sciences, Shenzhen, China. ✉e-mail: Bettina.Boettcher@uni-wuerzburg.de

portals in the vestibule that provide the cytoplasmic access to the pore. These lateral portals convey selectivity and restrict the conductance of some channels[7,9–12]. In YnaI, the portals are particularly small and limit the maximal conductance to only 0.1 nS compared to 1.3 nS in MscS. The arrangement of vestibule, pore and paddles creates hydrophobic pockets which extend into the cytosolic space, where they are populated by lipid molecules in an orientation almost parallel to the plane of membrane[7,8,13]. The TM paddles are in contact with the surrounding membrane and are therefore responsible for sensing the membrane tension, while the vestibule with the pore formed by the TM3 helices conveys conductivity and selectivity.

Opening of MscS-like channels is characterised by an outwards movement of the sensor paddles on the periplasmatic side, resulting in an increased cross-section in the membrane plane[14–16]. The paddle rearrangement pulls the pore helices apart and enables the passage of small solutes and water molecules between the vestibule and the periplasm. In the prototypic MscS, the paddle movement is concomitant with tilting of the whole paddle and relocation of the pore into the plane of the membrane. The movement of the paddles is accompanied by a reorientation of the lipids in the pocket and a transition from a curved local membrane environment into a planar bilayer arrangement in the open state. A hallmark of the tilting mechanism is a very long outer paddle helix, which extends into the cytosol and forms the entrance to the cytosolic pockets. In the closed state, these helices come together on the periplasmic side, resulting in a cone-shaped TM part, and trap the head group of a so-called hook or gatekeeper lipid in the centre of the membrane[17,18]. Alignment of lipid molecules in the membrane with the hook lipid induces the local curvature in the surrounding membrane[19].

Yet, compared to MscS, YnaI is lacking some of the structural key elements: In the closed state, the channel is less cone-shaped, and the paddles interdigitate with the lipids in the membrane without inducing marked local curvature. Furthermore, the outermost paddle helix is short and buried in the membrane, which makes tilting unlikely without distorting the local membrane environment. With electron cryo-microscopy and image processing, we show that the underlying mechanism for channel opening in YnaI is distinct from that of MscS. A chimera of both channels shows that the sensor paddles dictate the structural reorganisation of the TM part and the gating threshold, while the pore/vestibule module is important for conductivity. Furthermore, the residues of the pore define the structural rearrangement of the pore upon opening.

## Results

### Lipid supplementation stabilises YnaI in the closed state

Closed YnaI was purified for structure determination by electron cryo-microscopy in n-dodecyl-β-maltoside (DDM) supplemented with 0.5 mg/ml Azolectin (condition I+), resulting in a 2.2 Å map (Supplementary Figs S1–S3). The model built from residues 3–334 resembled the one of YnaI purified in LMNG by Hu et al.[8] with minor differences in the outermost TM helix (TM(−2)), which is frame shifted by 3 residues. In our model, Trp29 in TM(−2) is at the cytosolic interface of the membrane. This is the only tryptophan in the sensor paddle of YnaI, and it provides a typical side chain found at the membrane interface, where it tightly anchors the TM helix[20]. The better resolved paddle helices allowed us to assess the intra-paddle stabilisation. We found that each sensor paddle has four TM helices that are interconnected by 1-2 hydrogen bonds to the adjacent paddle helix. For example, Asn8 in TM(−2) makes a hydrogen bond with Tyr63 in TM(−1). This hydrogen bonding network suggests that the whole paddle forms a rigid entity within the membrane (Fig. 1a).

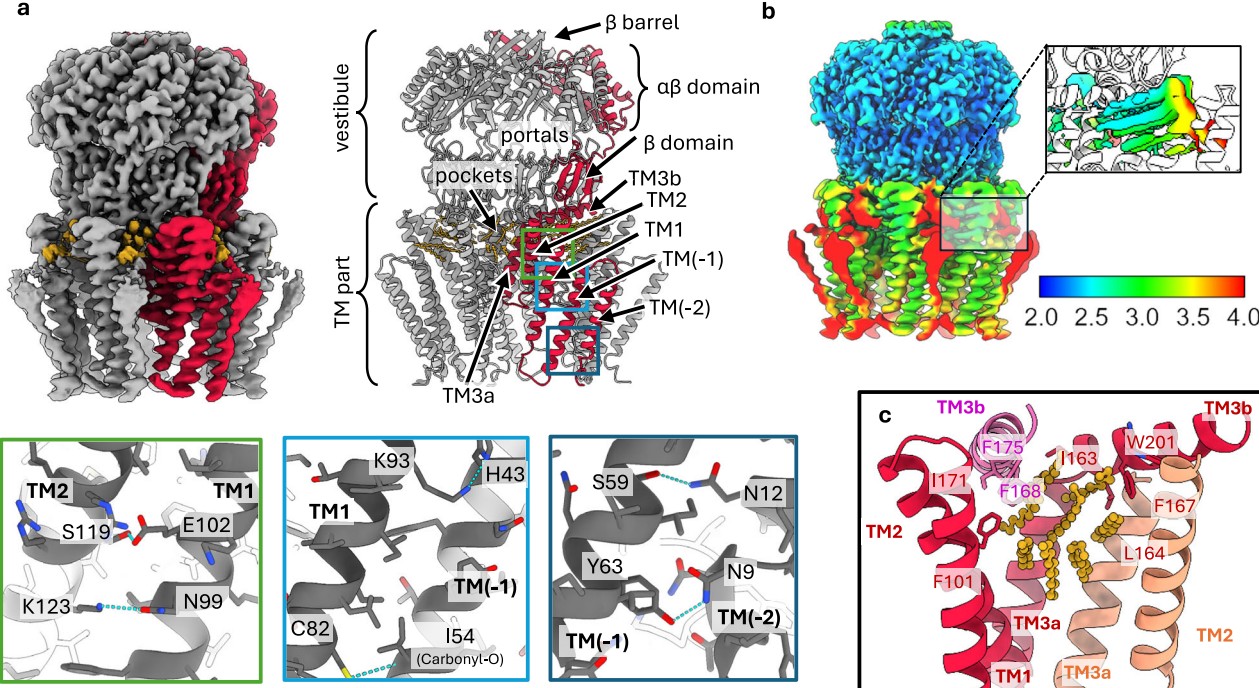

**Fig. 1 | Structure of closed YnaI. a** On the left, the cryo-EM map of closed YnaI acquired under condition I+ is shown in grey with one subunit highlighted in red and acyl chain densities in yellow. The right panel shows the corresponding model in the same orientation and colour code, and labels of relevant structural parts and domains. Close-ups of the green, yellow, and blue boxes are shown below and depict the intra-subunit hydrogen bonding network (blue dashed lines) between adjacent sensor paddle helices. **b** The map of closed YnaI is coloured according to the local resolution of the map. The colour key is shown on the right. The insert shows the model of YnaI in white with the density of the alkyl chains in the pockets coloured by the same local resolution key. **c** Only one pocket is depicted, and the helices and residues are coloured by subunit. One pocket is spanned by the helices TM1, TM2, TM3a, and TM3b from one subunit (red), TM2 and TM3a from one adjacent subunit (salmon), and TM3b from the adjacent subunit on the other side (pink). The eight modelled C$_{12}$ chains are shown in yellow.

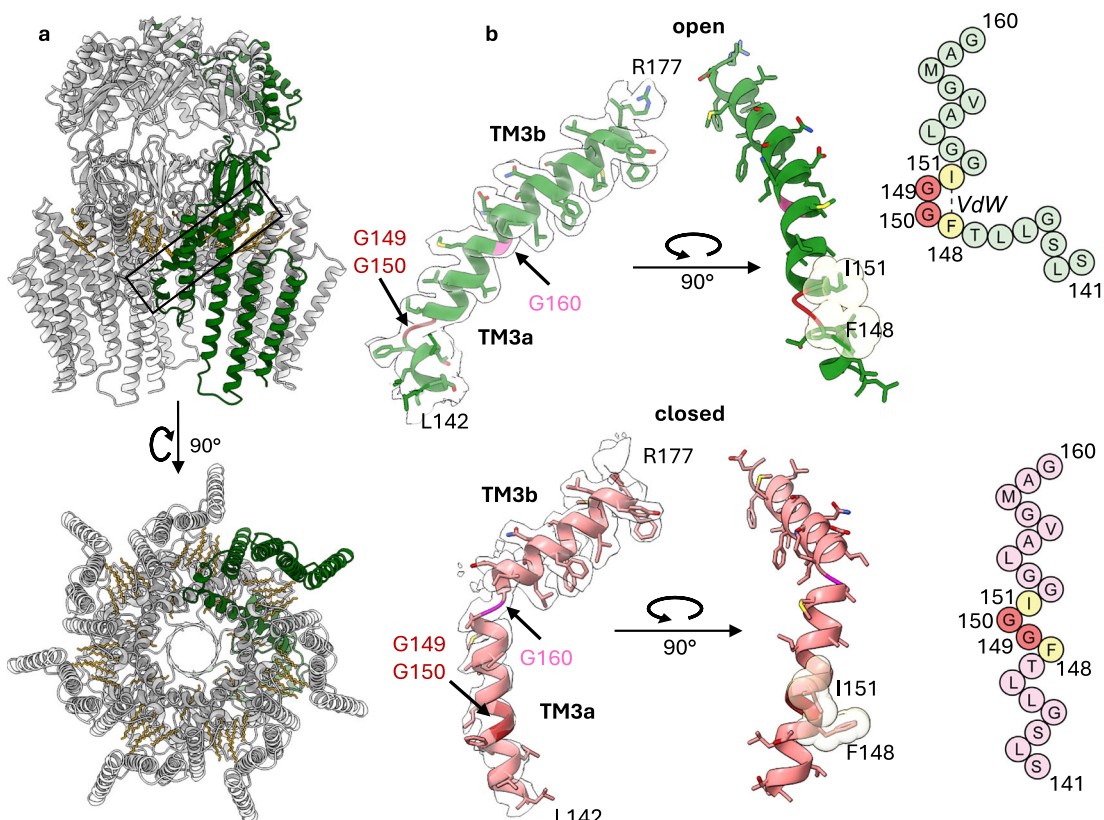

**Fig. 2 | Structure of open YnaI. a** The model of open WT-YnaI is shown in grey with one subunit in green. Alkyl chains are shown as yellow sticks. The pore helices TM3a/3b, indicated by the black box, are isolated shown in (**b**). The model of open YnaI is viewed from the periplasmic side along the pore axis, showing a wide pore. **b** At the top, the conjoined helix is shown in its corresponding density. The kink at Gly160 (pink), which divides the pore helices TM3a and TM3b in the closed conformation, diminishes, while a novel kink at Gly149-Gly150 (red) is created in the open conformation. Gly149-Gly150 are flanked by two large hydrophobic residues, Phe148 and Ile151, which interact with each other via Van der Waals bonds in the open state, stabilising the kink. The van der Waals radii are shown as transparent yellow spheres. On the right, a scheme of the helix is depicted. At the bottom, the helices TM3a and TM3b are shown for the closed conformation analogously to the open conformation.

The map of closed YnaI resolves the transmembrane part of YnaI better than previous maps (Fig. 1a, b and Supplementary Fig. S3)[7,8,12,13], suggesting structural stabilisation. We attribute this stabilisation to the added lipids, similar as observed previously for MscS[21]. In the lipid supplemented YnaI, we could model eight $C_{12}$ chains, representing alkyl chains, per subunit in the respective densities in the cytosolic pockets, yet we could not distinguish whether they were contributed by detergent or lipid. The eight alkyl chains formed parallel bundles with relative orientations that were compatible with four phospholipid molecules in a membrane context (Fig. 1b, c). The bundles were wedged between three adjacent subunits. The central of these three subunits generated most of the pocket with contributions from TM1, TM2 and TM3a/b. The other two subunits contributed either TM2 and TM3a or TM3b (Fig. 1c). TM3b formed van der Waals interactions with alkyl bundles in two adjacent pockets. While its Phe168 and Phe175 coordinated the alkyl chain in one pocket, Phe167 contacted the alkyl chain in the next bundle. Thus, the dispersed alkyl chain bundles were part of a hydrophobic inter-subunit interaction network that was located between the transmembrane part and the vestibule outside of the plane of the membrane. The pockets in the cytosol were continuous with the grooves between the paddles, implying that lipids can enter the pockets from the reservoir of the membrane.

## Opening YnaI

To generate an open conformation of YnaI, we introduced the mutation A155V (YnaI[A155V]). This mutation is analogous to A106V in MscS[14], A320V in MSL1 (*A. thaliana*) and G924S in MscK (*E. coli*)[15,16], which all show open pores in their respective structures. However, YnaI[A155V] purified with standard DDM concentrations produced the closed state (Supplementary Figs. S1 and S4). Also, the exchange of DDM to LMNG Lauryl Maltose Neopentyl Glycol (LMNG, condition II), which stabilises MscS most efficiently in the open state[21] (Supplementary Figs. S1 and S2), did not result in open WT-YnaI or open YnaI[A155V]. However, higher concentrations of DDM throughout the purification, which is sufficient to open MscS[21], also stabilised the majority of YnaI channels in the open state (Supplementary Figs. S1, S2, S5). Open WT-YnaI channels with an overall resolution of 2.3 Å, and the overall structure of open YnaI[A155V] were indistinguishable (Supplementary Fig. S4). This implicated that WT-YnaI and YnaI[A155V] adopted the same open conformation under the same purification conditions. The most prominent feature of open YnaI is the diminishing of the kink at Gly160, which separates the pore helices TM3a and 3b. In addition, on the periplasmic side the helix TM3a kinks outwards at Gly149-Gly150, resulting in a wide-open pore (Fig. 2). Because Trp29 anchors the outer paddle helix TM(−2) to the cytosolic leaflet, the pockets in the open state are pulled towards the plane of the cytosolic membrane leaflet and the pocket lipids align with the lipids of the cytosolic leaflet (Supplementary Fig. S6g).

Compared to closed YnaI (Fig. 3a and Supplementary Fig. S6), conductance is enabled by an increase in the pore diameter from ~9 Å to 22 Å in open YnaI and a rearrangement of the side chains. This widens the pore in the region of the sealing residues M158 and L154 (Supplementary Fig. S6b) and makes the interior of the pore less hydrophobic in the open state (Supplementary Fig. S6a). Notable

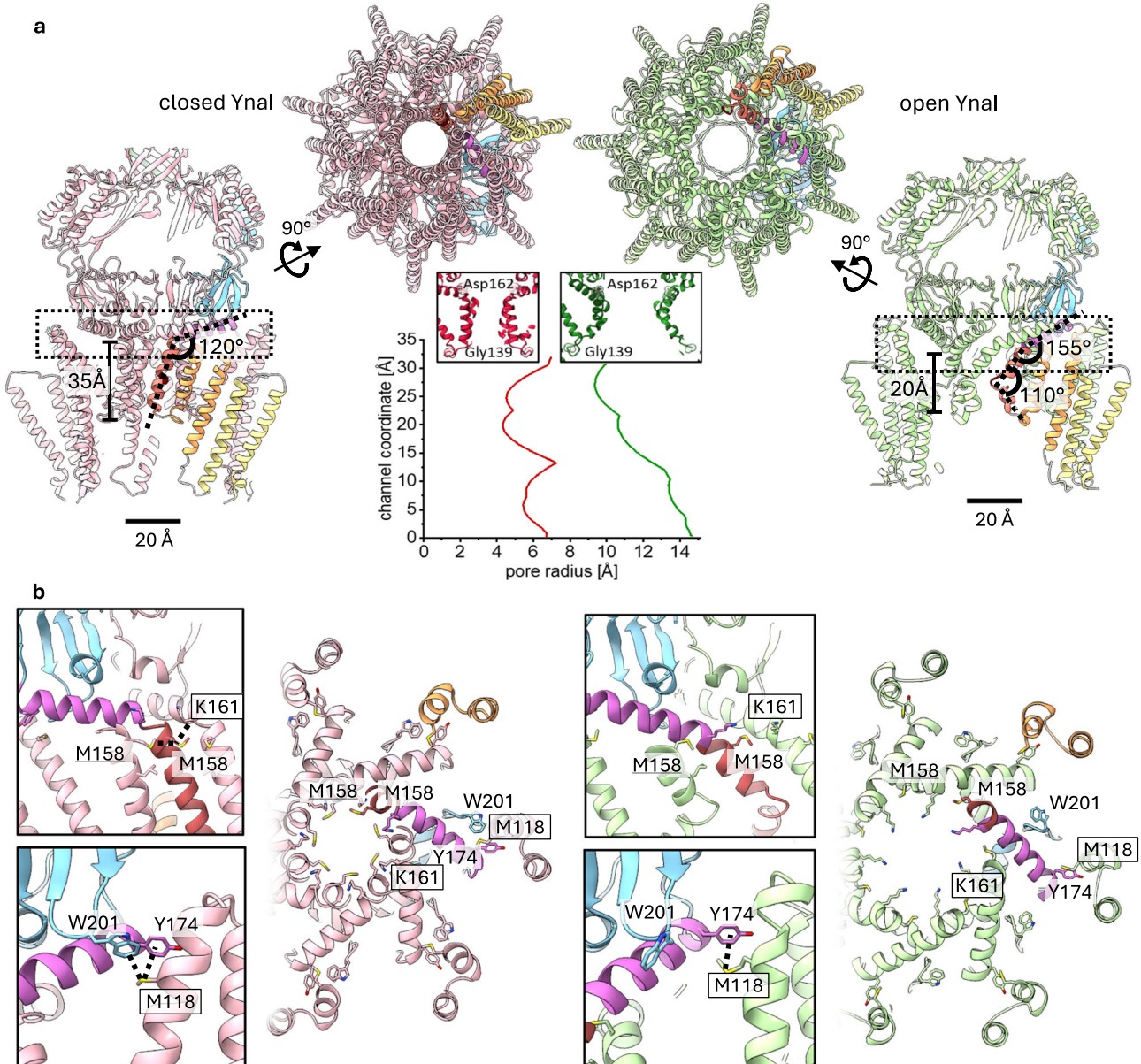

**Fig. 3 | Comparison of closed and open YnaI. a** Closed (light pink, left) and open (light green, right) YnaI are shown from the side and viewed along the pore from the periplasmic side. One subunit each is coloured according to their organisational units: helices TM(−2) and TM(−1) are coloured in yellow, and TM1 and TM2 in orange; helix TM3a in red and TM3b in purple, the β- and αβ domains are depicted in light blue, and the β barrel in green. The bar indicates the approximate length of the pore. The dotted box depicts the position of the slice shown in b. In the middle, the pore profiles determined with HOLE [47] of closed (red trace) and open (green trace) YnaI are shown. **b** The colour scheme is the same as in (**a**). A slice is shown that encloses the cytoplasmic part of the pore with relevant residues labelled.

Interactions are enlarged in the boxes. In the closed conformation, Met158 of one subunit can interact with the Met158 (underlined) of one adjacent subunit and Lys161 (boxed) of the other. Furthermore, a methionine aromatic bridge is present between Trp201 and Tyr174 of one subunit and Met118 (boxed) of a neighbouring subunit. In the closed conformation, Met158 is the diameter-constricting residue of the pore, while it is Lys161 in the open conformation. In the open conformation, Met158 can interact with neither residue of an adjacent subunit because of the wide-open pore. Also, the methionine aromatic bridge is broken, because Trp201 tilts away from the other involved residues.

hydrophobic residues are Leu142 and Leu146, which render the periplasmic side of the closed pore hydrophobic, are moved outwards in the open state because of the outwards kinking of TM3a (Supplementary Fig. S6b). In addition, the electrostatic potential of the pore is unevenly distributed (Supplementary Fig. S6c). The pore leads to an indentation at the periplasmic side, which is lined with negatively charged residues Glu66, Asp79, and Glu16 (Supplementary Fig. S6d).

## YnaI opening transition
The availability of the closed and the open conformation of YnaI allowed us a closer look at the underlying mechanism. Upon opening,

the gross architecture of the cytosolic vestibule stays largely the same, while the diameter of the TM part increases on the periplasmic side (Fig. 3a). In the open state, the pore is wider and shorter than in the closed state – the effective pore length is reduced from -35 Å to -20 Å. The increase in the pore diameter is concomitant with tilting of the pore helices, rejoining TM3a/3b into a long continuous helix, while shortening TM3a towards the periplasm by kinking it outward. Joining of TM3a and TM3b changes the angle between the helical backbones at their pivot point G160 from 120° in the closed conformation to 155° in the open state showing that the newly formed helical segment is still bent (Fig. 3a). The straightening of TM3a/3b is reminiscent of the

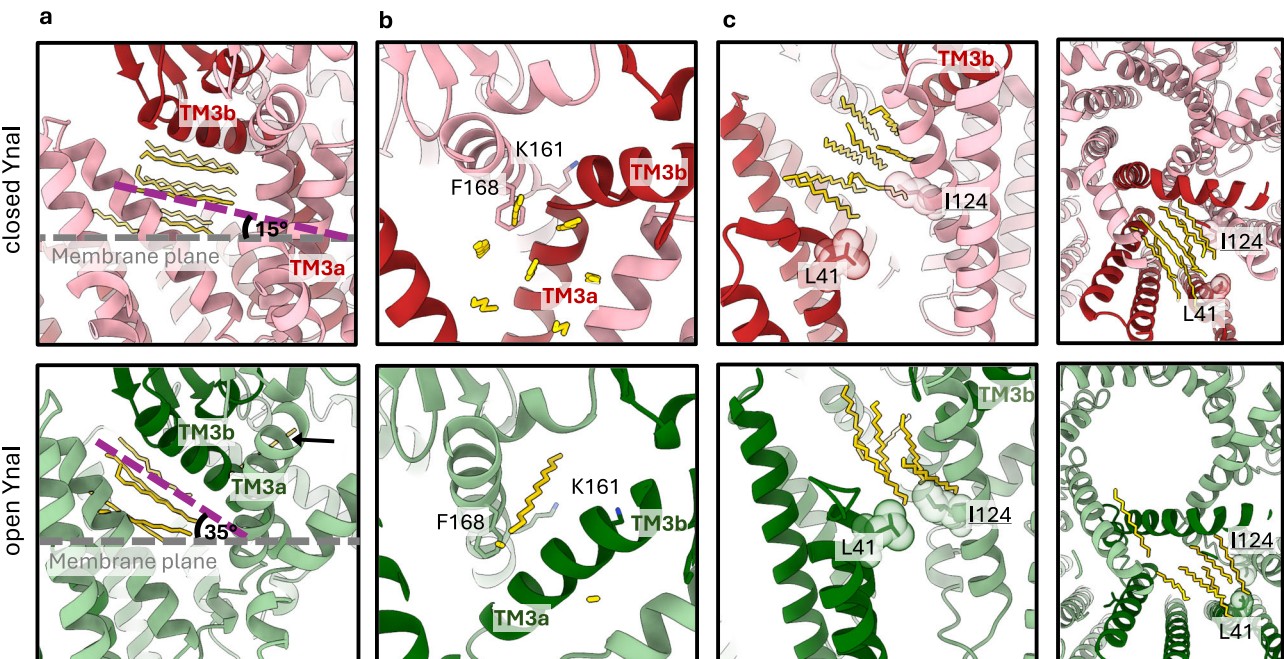

**Fig. 4 | Reorganisation of the alkyl chains in the pockets of closed and open YnaI.** The figure shows the modelled alkyl chains as yellow sticks of closed (top, red) and open (bottom, green) YnaI. One subunit each is highlighted in a darker colour. **a** The pocket lipids change their orientation as a bundle, and with respect to the membrane plane (grey dotted line), the angle is increased from 15° to 35°, following the change of the helix TM3b. **b** Upon opening, a gap arises at Phe168 due to the straightening of helix TM3a-3b. In the open conformation, one alkyl chain enters this gap, showing a different tilt than the other pocket lipids. This alkyl chain is also indicated by the black arrow in (**a**). **c** In the open state, a hydrophobic bridge is built between Leu41 of one subunit and Ile124 of a neighbouring subunit. Their van der Waals radii are depicted as transparent spheres. This bridge probably prevents the pocket lipids from leaving the channel in the open conformation and could play a role in closing the channel when membrane tension is released.

structural rearrangements of the pore helices of the large MscS-like channels MSL1 and MscK[15,16]. At the periplasmic side, the helical breakpoint in TM3a is located at Gly149 and Gly150, which are part of the [149]GGIGG[153] motif previously described[7], and the chain continues as a short helix from Leu146 until Ser141 at the periplasmic side of the pore entrance. The angle spanned between the two helix portions is 110° (Figs. 2b, 3a). We noticed that kinking at Gly149 and Gly150 enables the two large flanking residues, Phe148 and Ile151, to form van der Waals bonds that stabilise the kink in the open conformation (Fig. 2b). Kinking of TM3a moves the connected sensor paddle as an almost rigid body outward. In addition, the periplasmic ends of the helices tilt outwards by ~5° toward the cytosolic side, which gives the TM region a less tapered appearance in the open state (Fig. 3a). The diameter of the periplasmic side of the TM part increases from 67 Å in the closed state to 90 Å in the open state while the diameter at the cytosolic side remains unchanged (Supplementary Fig. S6e).

Next, we investigated the inter-subunit interactions that change between the open and closed conformation. In the closed but not open conformation, the pore lining Met158 interacts with Met158 of one adjacent subunit and with Lys161 of the other adjacent subunit. Similarly, Trp201 in the β domain interacts with Met118 at the cytoplasmic end of helix TM2 of the adjacent subunit in the closed state (Fig. 3b) but pivots away in the open state. In the outer TM region, Asn8 is near Lys71 of the neighbouring subunit, and forms an H-bond with it (Supplementary Fig. S6f) but loses the interaction in the open state, when the distance between these two residues increases to >15 Å. Comprehensively, upon opening, many inter-subunit interactions in the TM region of YnaI are lost.

In the maps of the open channel, we resolved six alkyl chains per pocket (Fig. 4a). We noticed that the arrangement of the alkyl chains between the open and closed state differs: With respect to the membrane plane, the angles of the chains change from approximately 15° in the closed conformation to 35° in the open conformation. One of the alkyl chain densities enters the vestibule close to Lys161 via the pockets in the open conformation. It intercalates between two adjacent pore helices and stabilises the larger pore diameter. The gap for the alkyl chain is generated by the straightening of helix TM3a-3b that retracts the Phe168 side chain in the helix TM3b, which would otherwise occlude the gap (Fig. 4b). This is consistent with the reported high lipid accessibility of Phe168[22]. Towards the plane of the membrane, the cytosolic pocket is closed by a hydrophobic barrier formed by Leu41 and Ile124 of neighbouring subunits. This barrier is within the cytosolic leaflet of the membrane and blocks the exchange of lipids between pockets and the membrane in the open conformation but not in the closed conformation (Fig. 4c).

## Open and closed conformations of a YnaI-MscS chimera

Our structural analyses of closed and open YnaI do not show molecular interactions between the paddle helices and the pore and vestibule. This suggests that the paddle (the sensing module) moves independently from the pore and vestibule (the conductivity module). To further investigate this hypothesis, we explored a chimeric construct comprising the sensing module of YnaI and the conductivity module of MscS. As the splitting point, we defined Gly139 of YnaI, or Gly90 of MscS, respectively. This glycine residue is conserved within the *E. coli* MscS-like family members (Fig. 5a). We have conducted electrophysiology experiments on *E. coli* MJF641 (Δ7) to identify the activity of the chimera (Fig. 5c, top). Due to the MscS part, the chimera exhibits a higher conductance than WT-YnaI (0.1 nS[4]) (Fig. 5b, c). However, in the chimera the ability of the channel to open stably is disturbed: it does not show the typical staircase-like opening behaviour as wildtype channels do, but it opens with a very short dwell time (*flickery*) instead (Fig. 5c and Supplementary Fig. S7a). Due to this behaviour, the time resolution is insufficient to identify the exact conductance for the fully open state. But at very high pressures, the current reaches a plateau at ~50 pA at +40 mV. This current is also confirmed by the Gaussian fit of

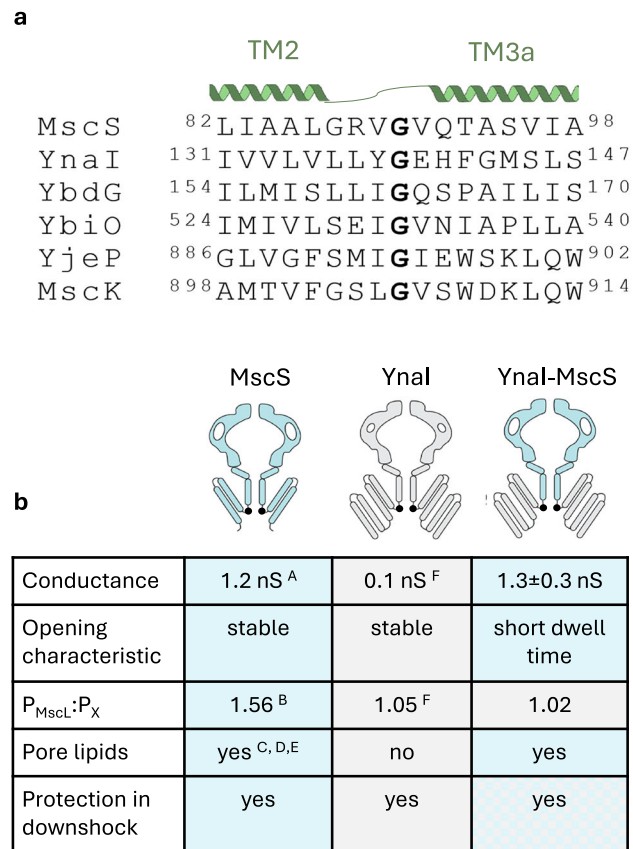

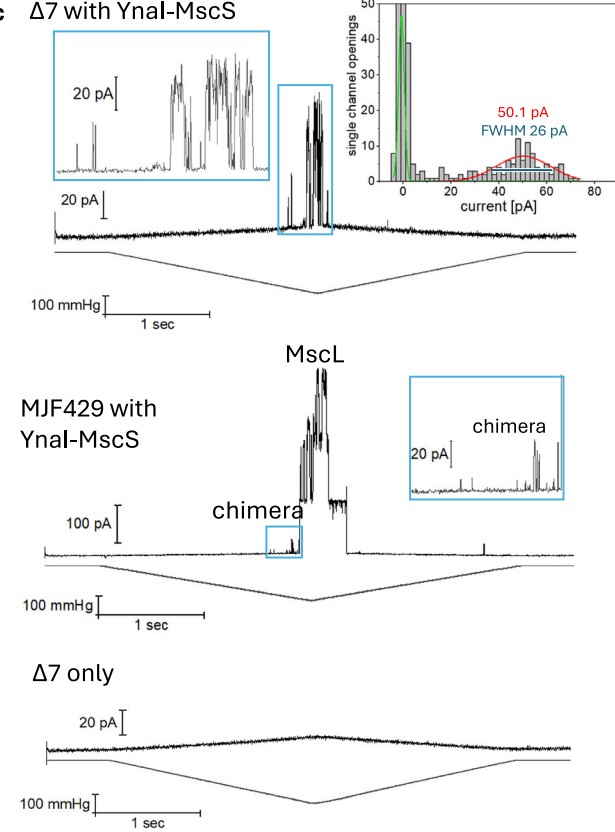

**Fig. 5 | Functional analysis of a YnaI-MscS chimera. a** The YnaI-MscS chimera was designed based on a glycine (bold) in the loop connecting the helices TM2 and TM3a. This glycine is highly conserved within the *E. coli* MscS channel family, which is Gly139 in YnaI (grey in right scheme), and Gly90 in MscS (blue in right scheme), respectively. **b** The YnaI-MscS chimera shows characteristics of both donor channels, YnaI and MscS, but also novel features, like the short open dwell time. **c** Representative current traces from membrane patches from *E. coli* MJF641 (ΔMscS, ΔMscL, ΔYnaI, ΔYbdG, ΔYbiO and ΔYjeP, ΔMscK (Δ7))- and MJF429 (ΔMscS, ΔMscK)- derived giant protoplasts expressing the YnaI-MscS chimera are shown. The inserts of the regions highlighted in the blue box are shown on the side.

For the recording of the Δ7-derived protoplasts expressing the chimera, the corresponding all-point amplitude histogram is shown on the right. The Gaussian fitting of the current-histogram (red curve) shows 50.1 ± 0.6 pA (maximum value ± standard error) and 26 pA (FWHM, blue line). This equals a conductance of 1.3 ± 0.3 nS (expected value ± sdev). The chimera opens close to the opening pressure of MscL and shows short dwell times (*flickery*). A trace from membrane patches from untransformed *E. coli* Δ7-derived giant protoplasts is also shown. No activities are visible. All electrophysiological experiments were conducted at +40 mV. Source data are provided as a source data file. References: A[2], B[23], C[17], D[25], E[21], F[4].

the corresponding all-point amplitude histograms (50.5 ± 0.6 pA (maximum value ± standard error) and 26 pA (FWHM) (Fig. 5c) and 50.1 ± 1.7 pA (maximum value ± standard error) and 22 pA (FWHM) (Supplementary Fig. S7a)). These currents refer to a conductance of 1.3 ± 0.3 nS (expected value ± sdev) for the open chimera and are similar to the conductance of MscS (1.2 nS[2]). We further conducted electrophysiology experiments on *E. coli* MJF429 for the investigation of the pressure threshold (Fig. 5c, middle), which showed that the YnaI-MscS chimera exhibits the high pressure required for opening, like YnaI (pressure threshold for opening $P_{MscL}$:$P_{YnaI-MscS}$ is similar to YnaI (1.02 ± 0.05 (YnaI-MscS chimera) vs. 1.05 (WT-YnaI[4]) vs. 1.56 (MscS[23])) (Fig. 5b). This emphasises that the required gating pressure is conferred by the sensor paddles. Although only the genes for the mechanosensitive channels MscS and MscK are eliminated in the *E. coli* MJF429 strain[4], we can distinguish the observed novel activities of the YnaI-MscS chimera from the endogenous channels YbdG, YnaI, YbiO and YjeP, because the combination of an MscS-like conductance and a required opening pressure similar to MscL is unique among these channels. The majority of above mentioned endogenous channels have no detectable (YbdG[5,24]) or a small conductance of only 0.1 or 0.3 nS (YnaI, YjeP[4]). YbiO, although having a conductance of 1 nS, opens at a pressure threshold $P_{MscL}$:$P_{YbiO}$ of 1.2[4,7] and is furthermore only scarcely expressed or assembled in the membrane[4]. However, as

the YnaI-MscS chimera opens almost at the same pressure required to open MscL, it is possible that we miss most of the chimera activities in the MJF429 strain, because it is overlaid by MscL. Untransformed *E. coli* MJF641 did not show any activities (Fig. 5c, bottom). The YnaI-MscS chimera is also functional in a cellular context and provides protection in hypoosmotic downshock assays (Supplementary Fig. S7b).

The structure of the chimera purified under condition I+ (Supplementary Fig. S8a–c) and analysed by cryo-EM (Supplementary Fig. S9a–d) represents a closed state that shows the MscS-derived conductivity module unchanged (Fig. 6a, b). The YnaI sensor paddles of the chimera have still the same structure as observed for WT-YnaI but are slightly shifted and tilted as a rigid body compared to it (Fig. 5a, b, d). As a result, the innermost paddle helix TM2 aligns with that of WT-MscS. We also observed the designated pore lipids of closed MscS[17,25], indicating that accumulation of lipids is a property of the MscS pore, probably related to its high hydrophobicity (Supplementary Fig. S9c). The helical backbones of the pore helices are highly similar in the chimera, YnaI, and MscS (Fig. 5b). The chimera purified under condition III (Supplementary Fig. S8a–c) and analysed by cryo-EM (Supplementary Fig. S9e–h) shows an open state with a novel pore architecture, in which we did not observe pore lipids (Fig. 5a, c, Supplementary Fig. S9g). Upon opening, the angle between helices TM3a and TM3b decreases from 135° to 100° (Supplementary Fig. S6a),

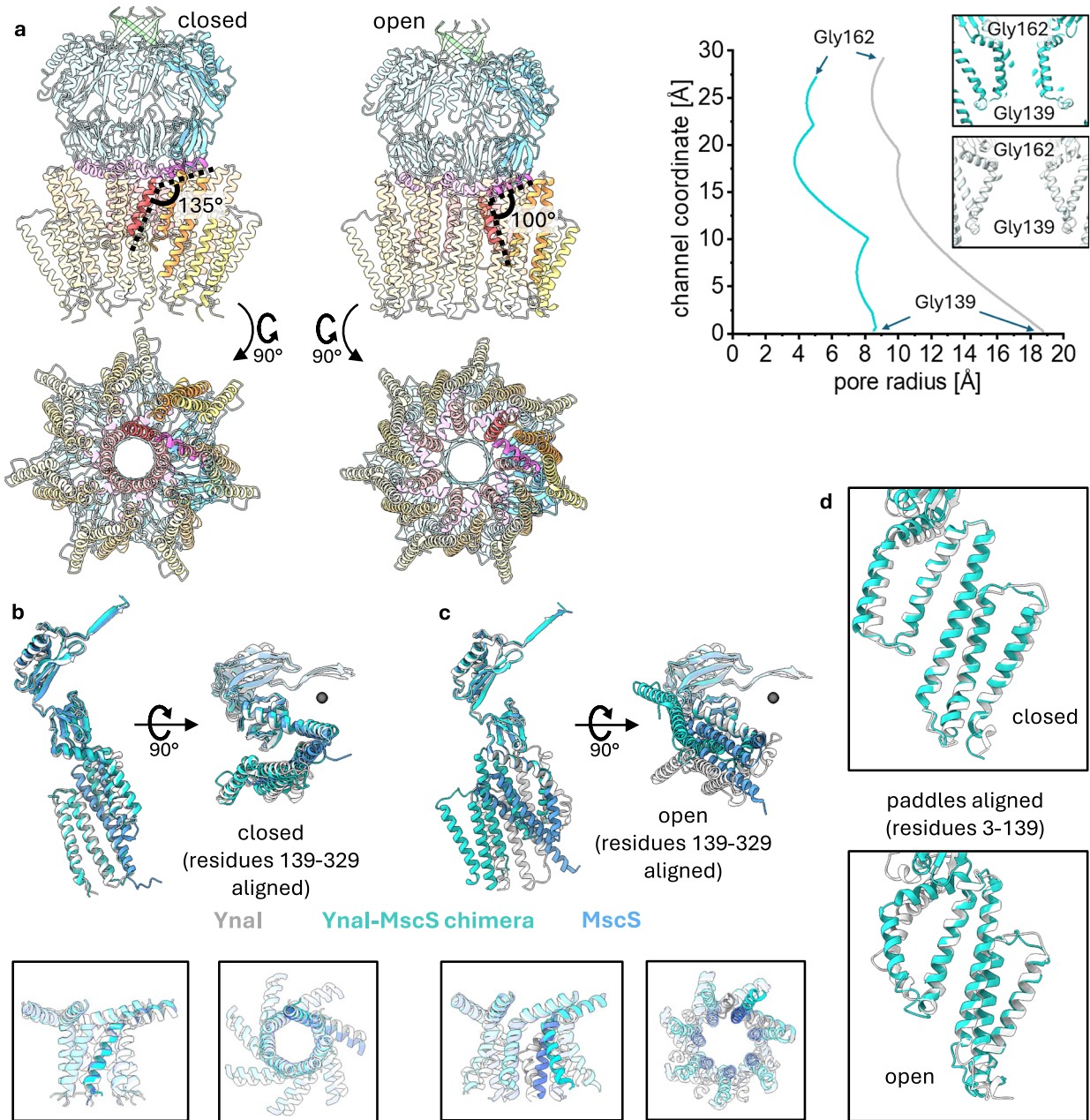

**Fig. 6 | Structural analysis of a YnaI-MscS chimera. a** The models based on cryo-EM maps are shown, coloured according to their organisational units: helices TM(−2) and TM(−1) are coloured in yellow, and TM1 and TM2 in orange; the helix TM3a in red and TM3b in purple, the β- and αβ domain are depicted in light blue, and the β barrel in green. One subunit is darker. While the architecture of the closed chimera largely resembles that of YnaI, it shows a novel pore arrangement in the open conformation. The helices TM3a are slightly bent outwards at the periplasmic side, resulting in a funnel-shaped pore. In addition, the paddle helices rotate clockwise in the open form – a feature that is observed for open MscS. The four paddle helices of one subunit remain a rigid body. On the right side, the pore radii determined with HOLE[47] are shown. The inserts show the pores with the residues G139 at the periplasmic side and G162 at the cytosolic side of the pore, which were chosen to determine the length to be considered for HOLE. **b, c** The pores and cytosolic domains of the closed (**b**) and open (**c**) structures are aligned. YnaI is shown in white, MscS in blue, and the chimera in cyan. One subunit is shown from the side (left) and the top (right). The black sphere marks the symmetry axis. The inserts in the black boxes represent the aligned pores (TM3a-TM3b) showing all seven subunits. **d** The paddles (residues 3–139) of the closed chimera and closed YnaI (top) and open chimera and open YnaI (bottom) are aligned, showing that the paddle organisation is essentially unchanged. For MscS representations, PDB 6RLD[18] (closed) and PDB 7ONJ[21] (open) were used.

which is more than in WT-MscS (135° to 120°). This leads to a funnel-like shape of the chimera pore and results in a unique positioning of the sensor paddles compared to the conductivity module, while they remain a rigid body (Fig. 6c, d). The outward movement of helix TM3a shifts them further clockwise (viewed from the periplasmic side) than in WT-MscS, but the paddles do not tilt within the plane of the membrane. Upon opening, the diameter of the TM part changes from 69 Å to 80 Å on the periplasmic side, while it remains unchanged at the

cytosolic side. The studies on the chimera exemplify that the sensing and conductivity modules function independently but adapt their conformational response to the overall channel organisation. The chimera showed that two modules derived from different channels resulted in a functional phenotype, with the gating threshold inherited from the sensor paddle donor, YnaI. The conductance is significantly higher than that of YnaI and similar to the pore donor, MscS. However, the flickery openings are neither inherited by MscS nor YnaI, as these

channels open stably. Noteworthy, the densities in the pockets of the closed and open chimera maps are only poorly resolved (Supplementary Fig. S9c, g). As the pocket architecture entailing lipid coordination involves the paddle helices and the helices TM3a and TM3b, the pockets in the chimera are built by helices of both donor channels. This could result in the observed disturbed lipid coordination, which in turn might result in the inability of the chimera to sustain the open conformation. Conclusively, we learned from this specific chimera 1.) that the behaviour of the kink dividing the helices TM3a and TM3b is a property inherent to these helices, 2.) that Gly149-Gly150 of YnaI are indeed necessary to kink the helix TM3a outwards, and 3.) that tilting of the paddles within the membrane plane is not related to the pore.

## Discussion

Knowledge about different conformational states of the MscS-like channels is inevitable for understanding the underlying molecular basis of mechanosensation. Our high-resolution structures of the larger MscS paralogue YnaI and a chimeric channel of YnaI and MscS allow the identification of shared and unique features of pore opening among MscS-like channels. A structural hallmark of opening MS channels based on the currently available structures is an area expansion of the transmembrane domain[15,16,21,25–27]. The force-from-lipids principle states that the anisotropic forces of the lipid bilayer[28,29] and their changes directly enforce channel gating[30–32]. In a membrane under tension, the channel conformation with a larger membrane cross-section is energetically more favourable. Hence, a simple in-plane area expansion can dispense the energy for conformational changes[33]. This can be described with Eq. (1):

$$\Delta G = \Delta G_{\sigma^{\rightarrow} = 0} - \sigma^{\rightarrow} \Delta A \tag{1}$$

Where $\Delta G_{\sigma^{\rightarrow} = 0}$ is the free energy difference for channel opening without tension applied, $\sigma^{\rightarrow}$ is the membrane tension, and $\Delta A$ the in-plane area change of the membrane cross section upon opening[34,35]. Furthermore, the sensitivity of an MS channel towards tension is given by (2).

$$\Delta A = \frac{d(\Delta G)}{d(\sigma^{\rightarrow})} \tag{2}$$

This means that a higher area expansion upon gating at the same tension infers a higher mechanosensitivity[34,35]. We noticed for YnaI that an area expansion only takes place at the periplasmic side of the TM part, while the area at the cytosolic side remains almost the same. The same is true for the MscS opening. But while the periplasmic in-plane area expansion for MscS is 22 nm², it is only 14 nm² for YnaI. YnaI opens at significantly higher membrane tension levels than MscS[4], which fits the observed different in-plane area expansions for MscS and YnaI. The YnaI-MscS chimera follows this observation, as its periplasmic in-plane area expansion is ~13 nm², correlating with a similarly high pressure required for opening.

Although area expansion seems to be the common basis for gating mechanosensitive MscS-like channels, there are further aspects to be considered. We could model numerous alkyl chains in our maps of closed and open YnaI. Though the origin of these alkyl chains could not be clearly attributed to either detergent or lipid molecules, their differing positions and orientations between the closed and open conformation are informative. This is in agreement with observations by Park et al.[19] based on coarse-grained MD simulations. These simulations indicated a re-orientation of the pocket lipid molecules of MscS and MSL1, stating that lipid reorganisation and not lipid removal accompany channel opening. Park et al. stated that membrane tension also favours an open channel conformation because the membrane perturbation in the pockets of the closed channel is energetically less favourable.

Our studies on a YnaI-MscS chimera indicated lipid coordination is an inherent feature and dependent on the molecular details of a channel. Specifically, we have observed that the presence of pore lipids is associated with the pore of MscS but not with the pore of YnaI, which has a less hydrophobic interior. Moreover, while the sensor paddles are donated solely from YnaI and the pore and vestibule solely from MscS, the hydrophobic pockets are mixed by helices of both donors. In our maps, the pocket ligand densities are hardly resolved compared to WT-YnaI and WT-MscS[17,18,21,25]. The mixed pocket architecture might not influence the number of coordinated lipids in the pockets, but suggests less tight coordination with a larger mobility of the lipids. Following the concept of Park et al.[19], this reduces the stabilising energetic contribution from reordering the pocket lipids in the open state. As a result, the energy barrier for reclosure is lower, leading to unstable openings as observed in the electrophysiological experiments.

Our open structure of YnaI shows the outwards bending of the pore helices TM3a at Gly149-Gly150, which divides TM3a into two sub-helices in the open conformation. The bending in this region has been investigated previously with two mutants: G149A/G152A, which should prevent kinking, was not able to protect cells in downshock assays, while G149P, which was designed to stabilise the kink, resulted in a gain-of-function phenotype that was reflected by a severe growth deficiency[7]. Outward bending of the pore helices TM3a in YnaI is, together with diminishment of the kink between the helices TM3a and TM3b, the major contributor to the significant pore shortening of open YnaI to approximately 20 Å. A shorter pore length means a reduction of the transmembrane diffusion distance and a lowering of the intra-pore diffusion energy barrier. Nonetheless, it is curious why YnaI opens to such a short pore with a large width, while the conductance is limited by the narrow lateral portals in the vestibule[7,8]. This suggests that in the open channel, ions can easily exchange between the vestibule and periplasm, while entering the vestibule from the cytosol remains restricted.

For the YnaI-MscS chimera, which inherited the pore helices TM3a and 3b and the cytosolic vestibule from MscS and therefore misses the di-glycine hinge of the YnaI pore, we did not observe the diminishing of the kink dividing the helices TM3a and TM3b and the outwards bending of the helix TM3a. Instead, pore shortening in the chimera is realised by reducing the kink angle, which gives rise to the observed funnel-shaped pore and is also accompanied by area expansion on the periplasmic side.

Comprehensively, our present study describes numerous structural features of YnaI, which are either unique or shared with other MscS-like channels. Lipids were postulated to initiate and control MscS gating[21,25,36]. The new findings show that pocket lipid removal cannot initiate gating in YnaI, as indicated by the fully crowded pocket, and highlight that area expansion following membrane tension is the major driving force for rearranging the protein scaffold towards a wider pore. Even more, we attribute a novel function to lipids, speculating that pocket lipid orientation might provide a mechanism for pore closing: When membrane tension is released, membrane lipids might push against the hydrophobic bridge built by Leu41 and Ile124. This results in a reorientation of the pocket lipids, which become more perpendicular to the membrane plane. The lipid tails then press against the pore and enforce kinking of the pore helices between the helices TM3a and 3b at Gly160 and thereby pore closure. Based on our high resolution YnaI maps, which allowed reliable model building of most of the closed and open channel, we suggest the following model for gating (Fig. 7): A tensed membrane enforces an area increase and shape change of the TM part because an expanded conformation is then energetically supported. Area expansion on the periplasmic side imposes an outward translocation of the sensor paddles, which induces bending of the connected pore helix TM3a at Gly149-Gly150. Concomitantly, the angle between TM3a and TM3b at Gly160

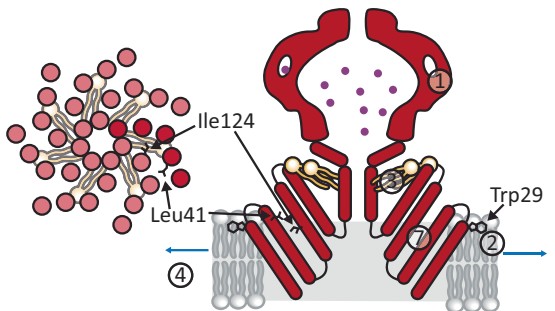 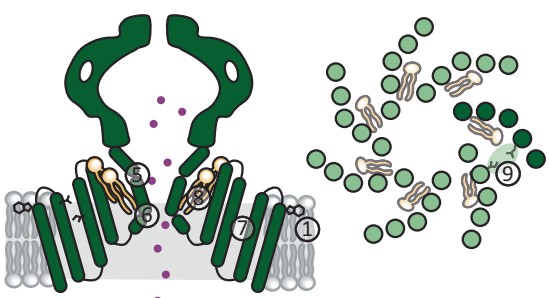

**Fig. 7 | The gating transition of YnaI.** YnaI is schematically shown in its closed (red), and open (green) conformation. Two subunits are shown for the side views, and the TM helices of all seven subunits viewed from the periplasmic side are shown at the very left and very right, respectively. In these top views, one subunit is coloured in a darker colour. In a model for gating YnaI, several aspects have to be considered. The portals (1) in YnaI are limiting for the conductance. The Trp29 (2) anchors the TM domain at the interface of the cytosolic membrane leaflet. In closed YnaI, lipids (3) fill the hydrophobic pockets. These lipids lie almost parallel to the membrane plane. Tension (4) leads to a then energetically favoured area increase of the periplasmic side of the TM part. The kink dividing the helices TM3a and TM3b diminishes (5), and the helix TM3a bends outwards (6), resulting in shortening of the pore. The paddles are relocated as a rigid body (7), and concomitantly, the pocket lipids (8) change their orientation to be more perpendicular with respect to the membrane plane. Radial relocation of the paddles enables the residues Leu41 and Ile124 to form a hydrophobic bridge in the open conformation (9), which separates the lipids in the pockets from the cytosolic membrane leaflet and reduces exchange with the lipid bilayer in the open state. In the open state, the squeezed pocket lipids press against the pore, ultimately enforcing its closure.

increases, leading to a decrease of the kink separating these helices, and as a result, the pore is opened. In YnaI, a hydrophobic bridge formed by Leu41 and Ile124 prevents the remaining lipids from leaving the pockets and can therefore not exchange with the membrane lipids in the open state. The rearranged open pore helices enforce a reorientation of the lipids, and the radial area that is occupied by lipids decreases upon pore opening. This is necessary because the diameter of the cytosolic side of the TM part merely increases upon opening. YnaI and the YnaI-MscS chimera highlight that although tension-induced area expansion is a common feature of mechanosensitive ion channels, there are certain features for fine-tuning their function.

## Methods

### Mutations

The pTrc_YnaI-LEH$_6$ construct[4] was used as a template to introduce the TEV cleavage site, ENLYFQS, by use of the Q5 site-directed mutagenesis kit (NEB). The YnaI$^{A155V}$ mutant was generated based on pTrc_YnaI-ENLYFQS-LEH$_6$ following the Stratagene QuikChange protocol using Q5 polymerase (NEB). Mutations were confirmed by sequencing (Microsynth Seqlab). The YnaI-MscS chimera synthetic construct with a C-terminal His$_6$ tag was purchased from Thermo Fisher Scientific and was subsequently cloned into a pTrc vector. The TEV cleavage site ENLYFQS was introduced using the Q5 site-directed mutagenesis kit (NEB).

### Expression and purification of YnaI constructs

Expression and purifications of the different YnaI constructs and applied purification approaches have been described previously for MscS[21] with minor modifications. The removal of the His-tag during purification prevented the formation of a disordered density beyond the β-barrel and allowed concentrating YnaI in the solubilised state. MJF641 cells (Δ7)[4] that overexpressed YnaI constructs with a C-terminal TEV cleavage site and a His$_6$ tag (LE-ENLYFQS-His$_6$) were resuspended and homogenised in solubilisation buffer containing either 1% (w/v) DDM (Glycon) (condition I and I+) or 1.5% (w/v) DDM (condition III) and incubated for 1 h on ice. After centrifugation, YnaI was filtered and applied onto prepacked 0.5 ml Ni-NTA agarose columns and washed with 40 ml washing buffer containing either 0.05% DDM (condition I and I+) or 0.5% DDM (condition III), 50 mM sodium phosphate buffer pH 7.5, 300 mM NaCl, 10% (w/v) glycerol, and 20 mM imidazole. For YnaI in LMNG, the washing buffer contained 0.03% LMNG instead of DDM (condition II). For elution, the same buffer was used but with 300 mM imidazole. The buffer of the protein containing

elution fraction was exchanged via PD10 columns (GE Healthcare) to cleavage buffer (0.05% DDM (condition I and I+) or 0.5% DDM (condition III), 50 mM sodium phosphate buffer pH 7.5, 300 mM NaCl, 10% (w/v) glycerol, 1 mM DTT) and TEV protease was added for 3 h. The samples were filtered and further purified on a Superose 6 10/300 size exclusion column (GE Healthcare) equilibrated with SEC buffer (50 mM HEPES pH 7.5, 150 mM NaCl, and 0.03% DDM or 0.02% LMNG, respectively. For closed YnaI with lipid supplementation (condition I+), the washing buffer contained 0.7 mg/ml Azolectin (Sigma) and the SEC buffer 0.5 mg/ml Azolectin. Peak fractions were concentrated using centrifugal filter units with a MW cutoff of 100 kDa (Amicon) to 4–6 mg/ml. Purifications were monitored via SDS-PAGE. For Western blot detection, a Penta-His HRP conjugate antibody (Qiagen) against the C-terminal His$_6$ tag of the constructs was employed.

### Cryo-EM

Quantifoil 300 mesh R 1.2/1.3 copper grids were glow-discharged in air for 2 min at medium power in a PDC-002 plasma cleaner (Harrick). Subsequently, 3.5 µl samples were applied on the grids and plunge frozen in liquid ethane using a Vitrobot IV (FEI/ Thermo Fisher) with following parameters: wait time 0 s, drain time 0 s, blot time 5 s, blot force + 25, chamber humidity set to 100 % and chamber temperature set to 4 °C.

For data acquisition, vitrified grids were transferred to a Thermo Fisher Titan Krios G3 transmission electron microscope. For the data sets of the closed and open WT-YnaI (condition I+ and III), and the closed and open YnaI-MscS chimera, movies were acquired at 300 kV with a Falcon 4i detector and a Selectris energy filter with a slit width of 5 eV at a nominal magnification of 130 kx, which corresponds to a calibrated pixel size of 0.946 Å. A total dose of approximately 40–70 e$^-$/Å$^2$ (Supplementary Table S1a, b) was used, and data was collected with the EPU-acquisition software (Thermo Fisher). For the data sets of YnaI$^{A155V}$ (condition II and III and WT-YnaI in LMNG (condition II)) movies were acquired at 300 kV with a Falcon 3 camera in integrating mode at a nominal magnification of 75 kx, which corresponds to a calibrated pixel size of 1.0635 Å. A total dose of 80–90 e$^-$/Å$^2$ was used.

### Single particle analysis

The cryo-EM structures were solved with cryoSPARC versions 4.1–4.4[37]. Movies were pre-processed in a cryoSPARC live session using patch motion and patch CTF for motion correction and dose weighting, and CTF estimation, respectively. Particles were picked using the blob

picker with a diameter of 124 Å and extracted with a box size of 256 px, followed by two to three rounds of 2D classification to remove junk particles. As initial volume for refinement, an ab initio reconstruction with C1 symmetry was generated for the initial refinements of the first WT-YnaI (condition I+). For other samples, previously obtained maps of YnaI were then used as initial models. The initial lowpass filter applied to the starting model was 30 Å. A heterogeneous refinement with four classes and applied C7 symmetry was conducted to distinguish between different populations. This step proved essential to differentiate between the closed and open conformations. Different classes from the heterogeneous refinement were subjected to individual non-uniform refinements with C7 symmetry and enabled global CTF refinement. For open WT-YnaI, a 3D variability analysis of the best class from the heterogeneous refinement was done, and the particles of the two best-performing clusters were joined for a final non-uniform refinement. Maps were sharpened with an applied B factor of −10 to −30. The remaining aligned particles, and the final map, of open WT-YnaI, and the closed and open YnaI-MscS chimera were additionally transferred from cryoSPARC to Relion 5[38,39] using pyEM[40]. In Relion, a 3D classification without blush regularisation was performed with the C7 symmetry relaxed. The best class(es) were 3D refined with blush regularisation and relaxed C7 symmetry. Details for the individual data sets are given in Supplementary Table S1a for closed and open YnaI and in Supplementary Table S1b for the closed and open YnaI-MscS chimera.

The detailed image processing workflows are depicted graphically for closed WT-YnaI (Supplementary Fig. S10), open WT-YnaI (Supplementary Fig. S11), the closed YnaI-MscS chimera (Supplementary Fig. S12), and the open YnaI-MscS chimera (Supplementary Fig. S13).

### Model building and refinement

All model building was performed in Coot[41] version 0.9.7, using existing structures as starting points (PDB 6ZYD (YnaI[7]), PDB 6RLD (MscS[18]). For most residues of the outermost helix TM(−2) of open YnaI, no clear side chains were visible, hence, the equivalent helices of the model of closed YnaI were placed in the density and fitted as a rigid body. For the open form of the YnaI-MscS chimera, no side chain densities were clearly attributable for the helices TM(−1) and TM(−2), therefore, also the helices were placed in the corresponding densities as rigid bodies. The nine C-terminal residues were resolved in neither of the maps. Phenix[42] version 1.17.1 was used for real-space refinement imposing secondary structure restraints, and subsequently, the models were validated with MolProbity[43]. Details for the individual data sets are given in Supplementary Table S1. All map- and model images were created with *UCSF Chimera* versions 1.15[44] or UCSF ChimeraX version 1.7.1[45,46].

The pore radii along the pore axis for closed and open YnaI were determined using HOLE[47] implemented in Coot[41] version 1.1.13.

### Electrophysiology

Giant *E. coli* protoplasts were generated as described previously[48]. Patch-clamp recordings were conducted on membrane patches derived from giant protoplasts using the strains MJF429 (ΔmscS, ΔmscK) and MJF641 (ΔmscS, ΔmscK, ΔynaI, ΔybdG, ΔybiO, ΔyjeP, ΔmscL) transformed with plasmids for overexpressing the YnaI-MscS chimera[2,4], and on untransformed MJF641 protoplasts. The cultures were induced with 1 mM IPTG for 1.5 h before protoplast generation. Excised inside-out patches were measured at +40 mV with identical pipette and bath solutions (200 mM KCl, 90 mM MgCl₂, 10 mM CaCl₂, 5 mM HEPES buffer at pH 7.0 adjusted with KOH). Data was acquired at 3 kHz filtration using an EPC-8 amplifier and the Patchmaster software (HEKA). Symmetric pressure ramps were applied over 4 s with a maximum pressure of 290 mmHg using a HSPC-1 high-speed pressure clamp system (ALA Scientific Instruments). The pressure threshold for activation of the YnaI-MscS chimeric channels was referenced against the activation threshold of MscL ($P_{MscL}$:$P_{YnaI-MscS}$) to determine the

pressure ratio for gating[49]. Reference measurements resulted in $P_{MscL}$:$P_{YnaI-MscS} = 1.02 \pm 0.05$ ($n = 9$; derived from three independent protoplast preparations from three different transformations). For the all-point amplitude histograms of a corresponding trace, a background trace resulting from measurements of the MJF641 strain only with the same pressure profile was subtracted before to have the baseline corrected. The histogram bars are depicted with a binning of 2 pA. Gaussian fitting was performed, resulting in two Gaussian curves, of which the one covering the currents up to ~15 pA is derived from noise. The maxima of the Gaussian distributions are quoted together with the standard error of the fit. The width of the Gaussian distribution was determined as full width at half maximum (FWHM). This value was converted into the standard deviation of the distribution (sdev = FWHM/2.355).

Values (conductance, gating threshold) for MscS and YnaI were taken from the literatures[4,7,50].

### Downshock assay

The hypoosmotic downshock assay was carried out as described previously[51]. *E. coli* MJF641 (Δ7) cells were transformed with the constructs of MscS, YnaI, or the YnaI-MscS chimera and grown in LB medium with an extra 0.3 M NaCl until an $OD_{600}$ of 0.28-0.3. As a control, untransformed Δ7 cells were grown. Expression was induced with 0.3 mM IPTG during the whole main culture. Cells were shocked by 20 times dilution into LB medium or as a control into LB with 0.3 M NaCl and incubated at 37 °C for 10 min. Subsequently, serial dilutions were plated onto plates with the same osmolarity (shock and control plates) and incubated at 37 °C overnight. The next day, colonies were counted and the survival ratio determined. Triplicates were performed, and the mean ratios and standard deviation were determined.

### Reporting summary

Further information on research design is available in the Nature Portfolio Reporting Summary linked to this article.

## Data availability

The EM maps of C7 processed closed and open YnaI (purified under condition I+ and condition III), and of the closed and open YnaI-MscS chimera (purified under condition I+ and condition III) obtained in cryoSPARC have been deposited in the Electron Microscopy Data Bank (EMDB) and corresponding atomic models have been deposited in the Protein Data Bank under following accession codes PDB 9H95 and EMD-51953 (closed YnaI), PDB 9H2P and EMD-51813 (open YnaI), PDB 9H2S and EMD-51817 (closed chimera), and PDB 9H2V and EMD-51818 (open chimera). Maps processed in Relion with C7-relaxed symmetry of open YnaI, and the closed and open YnaI-MscS chimera have been deposited to the EMDB under the accession codes EMD-51897 (open YnaI), EMD-51898 (closed chimera), and EMD-51954 (open chimera). For the preparation of Fig. 6, PDB 6RLD and PDB 7ONJ were used. Source data are provided in this paper.

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

## Acknowledgements

We thank Christian Kraft for technical assistance. Cryo-electron microscopy was carried out in the cryo-EM facility of the Julius-Maximilians-Universität Würzburg, funded by the Deutsche Forschungsgemeinschaft (DFG, German Research Foundation – Projects INST 93/903-1 #359471283, INST 93/1042-1 #456578072, INST 93/1143-1 #525040890). B.B. acknowledges project funding by the Deutsche Forschungsgemeinschaft (DFG, German Research Foundation - Projects 343886090, 538122946. The cryo EM-facility of the Julius-Maximilians-Universität Würzburg received funding from the DFG – Projects 359471283, 456578072, 525040890).

## Author contributions

V.J.F. performed biochemical preparations, V.J.F., T.R., and B.B. analysed cryo-EM data. V.J.F. and A.R. conducted electrophysiology experiments, with supervision for R.H. B.B. conceived funding and supervised the work. V.J.F. prepared the manuscript with contributions from all authors. All authors edited the manuscript.

## Funding

## Competing interests

The authors declare no competing interests.

## Additional information

**Supplementary information** The online version contains Supplementary Material available at https://doi.org/10.1038/s41467-025-63253-0.

