## [Transparent Peer Review file · Nature Communications]

Mechanosensitive Channel Engineering: A Study on the Mixing and Matching of Ynal and MscS Sensor Paddles and Pores

Corresponding Author: Professor Bettina Böttcher

Version 0:

Reviewer comments:

Reviewer #1

(Remarks to the Author)

Although mechanosensitive ion channels are found in both prokaryotes and eukaryotes, they exhibit significant structural diversity. Despite recent advances in understanding their working mechanisms and the detailed dissection of several key mechanosensitive ion channels (such as MscS, K2P, OSCA, et al.), the commonalities and diversities in the features of the inductive forces of these proteins still require further elucidation through additional examples. In the manuscript NCOMMS-24-71655 titled "Ynal exemplifies the diversity of structural gating mechanisms in mechanosensitive channels of small conductance," Vanessa J. Flegler et al. report the high resolution open and closed structures of Ynal and its chimera with MscS. Their findings show that Ynal opens through the radial relocation of the transmembrane sensor paddles, accompanied by a shortening of the pore, which contrasts with the mechanism observed in MscS. Interestingly, a thinner pore is also noted in the recently published OSCA channel (Yuanyue Shan et al doi:10.1038/s41467-024-51800-0). Additionally, the relocation of several lipids and the kink in the transmembrane region at a flexible residue are also reminiscent of the working mechanism of the OSCA channel. Also, the detergent or lipid environment largely shapes the state of Ynal. These results seem to provide clues to the commonalities between the MscS and OSCA families. Moreover, based on the functional data of the chimera containing both Ynal sensor paddles and the pore region from the C-terminal part of MscS, the authors propose that elements with different types of structural rearrangements can be mixed and matched within a single channel, as long as they support the common area expansion on the periplasmic side. Overall, the results are intriguing, and the manuscript is well-written. The only shortcoming is that the existence of the open and closed structures of several MscS channel somewhat diminishes its novelty. I am happy to recommend the publication of the paper after the following comment is addressed.

Major/minor comments:

1. Since the importance of the flexible residues Gly149-Gly150, it is suggested to provide functional experimental evidence of their role in channel opening.
2. Figure 6 and Figure S6b: There is no mention of an interaction between Leu71 and Ile124 in the previous structural analysis. It is unclear why this interaction is shown in the gating transition schematic in Figure 6. Additionally, Figure S6b shows the hydrogen bond between Lys71 and Asn8 disappearing in the open state of Ynal. This raises the question: Is residue 71 Leu or Lys?
3. SDS-PAGE in Figure S2b and Figure S4b: It is suggested to rerun the gels. The SDS-PAGE in Figure S7a mentions that the chimera expression level is low, but the bands are barely visible in the gel. The gel should be rerun and the image replaced accordingly.
4. Cryo-EM Analysis: It would be beneficial to include the detailed Cryo-EM analysis process for the closed and open Ynal-MscS chimera in the supplementary material.

Reviewer #2

(Remarks to the Author)

Ynal exemplifies the diversity of structural gating mechanisms in mechanosensitive channels of small conductance

The manuscript by V.J. Flegler et al reports high-resolution cryo-EM structures of the bacterial mechanosensitive channel YnaI both in the closed and open states, as well as the closed and open states of YnaI-MscS chimeric channel, containing a pore-lining helix and a complete cytoplasmic cage of MscS combined with four first TM helices of YnaI. The authors highlight the unique arrangement of WT YnaI channel, which, unlike closed MscS channel, does not produce a significant membrane invagination and has a wide funnel-shaped entrance with an unusually short conductive pore when opened. Based on the closed and open structures of WT and gain-of-function A155V mutant, the authors hypothesize on a possible gating mechanism of this channel. Having obtained closed and open structures of YnaI-MscS chimeric channel, they further speculate that the pore-lining helix and C-terminus on one hand and the sensor paddle on the other have separable functions: the pore helix and C-terminus control conductance, while the sensor paddle is responsible for tension sensitivity. Although cryo-EM structures of WT YnaI are not new, the data is of high quality, and allows to identify previously overlooked inter-molecular interactions within the oligomeric channel. However, the authors do not comment a lot on the differences between new and previously published structures (including those obtained by them earlier). The closed and open structures of YnaI-MscS chimeric channel, presented in the manuscript, illustrate an interesting, though not novel, concept of separable functions of different parts of the channel-forming protein, which is potentially applicable to many, not necessarily related, channels.

Unfortunately, in my opinion, the manuscript has a major problem, related to the functional data, because of which the conclusions presented by the authors are not justified enough. This fact does not allow me to recommend it for publication without serious modifications. The main concerns are summarized below in points #1 and, especially, #2.

1. Protein isolation and purification.

- Cryo-EM: I guess the construct included a GFP C-terminal tag, which was cleaved later, but it has not been mentioned either in the Methods or in the figure legends. In all size-exclusion chromatograms the amplitudes of the target peak P1 at ~15ml (WT YnaI on Fig.S2, YnaI-A155V on Fig.S4, YnaI-MscS chimera on Fig.S8) seem to be similar (about 1000 mAU), which indicates comparable amounts of protein.
- However, the whole cell Western blot (Fig.S7a) shows very low (if any at all) amount of YnaI-MscS chimera protein (indicated by an arrow). According to the description, the same cell line (MJF641) was used for protein production for both cryo-EM and electrophysiology. The authors mention only one vector (pTrc) in the Methods section, so I suppose it was also used in the both assays. Therefore, one would expect similar amounts of protein produced and, therefore, similar intensities of the bands. This obviously is not the case. In fact, the quality of the gel seems to me insufficient to tell if there is a real band for the chimera protein on Fig.S7a. The authors did not mention the antibodies they used either (that were probably anti-6His).

Also, why the chimera lane is separated from the rest ($\Delta 7$, MscS, YnaI, and the ladder) of the lanes? Was it obtained separately, under different conditions, on a different day, etc and then combined with an image of another gel? The authors should comment on this.

2. The concern about the presence of YnaI-MscS chimeric channel in the membranes of E. coli spheroplasts used for patch-clamping, raised by the reasons listed in the previous point, is very serious also because of the following:

- The authors state, that (lines 291-292), that “the chimera provides protection in hypoosmotic shock assay, despite being only weakly overexpressed (figure s7a,b)”. The same statement is also re-iterated in Figure 5a (“Protection in downshock” in the table). However neither in Figure 7, nor anywhere else in the manuscript there are data or references supporting this statement. This also applies to MscS and YnaI (Figure 5a, table).
- The same table indicates unitary conductance of YnaI-MscS chimera of ~1nS, which is about 3 times lower than unitary conductance of MscL under the same conditions (about 3.3 nS), however Figure 5b demonstrates “chimera” unitary currents of about 1/10 (or even less) of those of MscL, which clearly contradicts the data presented in the table.
- This also applies to Figure S7b and its legend. The channel openings labeled as YnaI-MscS (“#”) definitely have the conductance much lower than 1nS, because the openings of WT MscL (“**”) should be about 3nS.
- The reasons why the authors attribute two obviously different types of openings (very flickery and unstable in the upper and lower panels of Figure 5b and more stable table-like opening in lower panel of the same figure) are also unclear. It is not uncommon to see spontaneous low amplitude flickery activity when membrane is subject to near-lytic tensions, activating MscL (which is also present in the figure).
- Figure 5b and Figure S7b legends indicate the patch-clamp experiments were done in MJF429 (Δ MscK, Δ MscS) cell line, which still expresses endogenous MscL, YnaI itself (!), and three more (!) endogenous mechanosensitive channels of E. coli. Taking into account this background and the absence of ~1nS conductances presented in Figures 5b/S7b, it is more than likely that the electric activities, presented in these figures are not caused by YnaI-MscS chimera.
- Channel adaptation (Figure 5a table) is not illustrated/supported by experimental data; if taken from the literature – no references are provided.

Summarizing the arguments presented in points #1 and #2: all the functional data on YnaI-MscS chimera, presented in the manuscript seems to be mostly irrelevant and therefore can't be used for justification of the proposed YnaI gating mechanism (as stated, for example, in the two last sentences of the Abstract, lines 26-32). Hence, a substantial part of the manuscript's conclusions remain unsupported (by experimental data) and represent rather speculations than established facts.

Minor:

1. Lines 42-43: “...but release only small solutes with a very small flux”. That is not completely true, as, for example, WT MscS, which is naturally expressed at quite density, has a unitary conductance of ~1.2nS, which is only 3 times lower than the conductance of MscL. Therefore the fluxes mediated by MscS are rather big.
2. Lines 49-51: would be nice to have at least a reference here.

3. Lines 79-80. This point of view is quite outdated. For example, the recent publications of structures of MscS-related plant mechanosensitive channels FLYC1 and MSL10 indicate the lack of substantial membrane deformations and rather mild conformational changes during gating of these channels (Jojoa-Cluz et al, 2022; Zhang et al, 2023). Obviously, they very unlikely to share the same gating mechanism with bacterial MscS.
4. Lines 86-87. As mentioned above, this statement is not confirmed by experimental data.
5. WT Ynal structures, although are of high resolution and quality, are not new. The authors unfortunately do not discuss the advantages of and new details provided by these newly obtained structures.
6. Figure 1b. The resolution map, presented as an overlay of pseudo-colored density map and the protein structure is a bit difficult to comprehend. I'd suggest to take the usual approach and to demonstrate the pseudo-colored 3D density map of the whole channel instead.
7. The authors do not provide any figures illustrating the water-accessible pore profiles (obtained with HOLE or similar software) for any of the structures. It makes it difficult to appreciate the structural rearrangements, resulting in the channel opening for example. Also (as in line 175: "...increase in diameter from ~9A to 22A in open Ynal...") the references to the structural rearrangement are quite obscure as they are neither illustrated in figures, nor they specify the exact residues where the measurements were taken. This also applies to several other statements, such as, for example, found in lines 223-224: "...the effective pore length is reduced from ~35 A to ~20 A." There is no panel in Figure 3 illustrating this, there is also no direct indication between which residues this distance was measured.
8. Line 286: pressure ratio of 1.04 seems to be mistakenly labeled as "MscS". Also, the Methods only describe how the pressure ratio of MscL:chimera was obtained. What is the origin of the pressure ratios of MscL:Ynal and MscL:MscS? Were they measured by the authors or taken from the literature? In the latter case, the references should be presented.
9. There are a few sentences in the manuscript that look to me like containing duplicate, wrong, or missing words (lines 142, 178, 253-254, 558).

Version 1:

Reviewer comments:

Reviewer #1

(Remarks to the Author)

The authors have revised most part of my concerns. I have no additional suggestions.

Reviewer #2

(Remarks to the Author)

Major:

1. Reading the manuscript, I am still struggling to find a comprehensive discussion of the point outlined in the title, namely, "the diversity of gating mechanisms in mechanosensitive channels of small conductance". I would suggest the authors either to change the title to reflect the main points of the manuscript or to provide a more detailed review of MscS-like channels gating mechanisms.

2. In the Ynal closed state the authors identify 8 alkyl chains per subunit (lines 113-120). In the same time, in the open state only six alkyl chains per pocket (line 218) are identified, which suggests that one lipid molecule leaves the pocket upon/during channel opening. This observation contradicts the authors speculation that the pocket lipids cannot leave the pocket (or exchange with the lipid bulk of the membrane), as stated in lines 407-409. On the other hand, the authors state, that a) "the pockets in the cytosol were continuous with the grooves between the paddles implying that lipids can enter the pockets from the reservoir of the membrane" (lines 127-129) and b) the (membrane) "lipids might push against the bridge and become continuous with the pocket lipids" (lines 396-399), which is very confusing. I would suggest the authors to review the gating model description to make it more comprehensive.

3. In the Results section the authors do not comment on the localization of the lipid pockets in Ynal-MscS chimera (closed or open), neither they mention what lipid-like densities were identified in the pockets. This is very strange as the role of lipids in MscS and MscS-like channels gating is one of the main points of the manuscript.

Electrophysiology of Ynal-MscS chimera:

1. Open and closed conformations of a Ynal-MscS chimera. I'd suggest to start this part with the experiments done in MJF641 cell line and then to move to MJF429, as the previously unknown (Ynal-MscS chimera) unitary channel properties are characterized on the clean background of MJF641 first. Then, based on these results, it is possible to discriminate Ynal-MscS from the rest of potential endogenous MscS-like activities in MJF429.

Also, please provide the rationale why the membrane stretch-activated currents in MJF429 may not (or may) be caused by endogenous mechanosensitive channels, like, for example, YbiO, having similar to MscS unitary conductance, and lower tension sensitivity (doi: 10.4161/chan.20998).

I would also recommend to present the all-point histogram for figure s7 (and maybe figure 5c) raw trace data to illustrate the most probable conductance level(s) of Ynal-MscS chimera.

Minor:

Line 141: "...WT-Ynal and Ynal155 were indistinguishable..." On figure s4d, probably due to the graphical artifact (depth occlusion?) blue TM1 does not seem to be overlaid with a green one; maybe RMSD value will be more suitable here.

Lines 146-148: "the pockets in the open state are pulled into the plane of the cytosolic membrane leaflet and the pocket lipids align with the lipids of the cytosolic leaflet". I'd suggest to illustrate this conformational change with a figure (and to estimate the displacement in Å) as this feature is important for the further discussion of Ynal gating.

Figure 3: the pore profiles plot is erroneously labeled as figure 3b

Figure 4: please label the alkyl chains with a brighter color, as they are really difficult to see.

Line 213: "figure 3f" – there is no such figure in the manuscript.

Lines 215-217: "most inter-subunit interactions in the TM region of Ynal are lost" – I guess, the authors mean the interactions between the paddles of the subunits? I guess the TM3-TM3 interactions should be preserved, otherwise the pore would become unstable.

Lines 303-304: "However, the flickery openings are neither inherited by MscS nor Ynal" and are probably due to what? Maybe some speculations/hypothesizing on the origin of such a behavior?

Figure 6a: I'd suggest to add WT MscS (open and closed conformations) pore profile to the plot alongside with the Ynal-MscS chimera's. Could the differences give a hint of the reasons why the chimera activity is different from WT MscS?

Lines 360-363: Maybe I am missing something, but I do not see any description or figures illustrating hydrophobic pockets (and lipid-like densities in them) of the Ynal-MscS in Results.

Lines 379-380: "reminiscent of K⁺ channels, which only have a relevant pore length of 12Å". I do not think it is a valid comparison. The authors refer to the length of highly conserved selectivity filter, however the pores of potassium channels are far more complex and demonstrate constrictions, that in many cases are almost as narrow as their selectivity filters, of which Kir2.X channels (<https://doi.org/10.1126/sciadv.abq8489>) are a good example. Although the authors' reference might be true for some potassium channels, in general their pores cannot be approximated by a simple funnel shape, similar to that the authors suggest for Ynal. Also, as noted by the authors, Ynal ion selectivity (or rather ion preference), unlike potassium channels, is dictated by the side portals of its cytoplasmic cage, but not by the constrictions of the transmembrane pore.

Lines 385-386: "...cannot reduce the kink dividing the helices TM3a and TM3b and cannot bend the helix TM3a outwards". What is the force preventing this conformational change? The authors postulate independent functions of the pore and the paddle domains, so the MscS-derived pore of the chimeric channel is probably expected to adopt "open-MscS" conformation?

Miscellaneous:

Line 22: Despite several structures...

Line 93: ...and it(?) provides...

Line 157: ...which interact with each other(?) via Van der Waals...

Line 367: unstable

Line 388: which gives rise

Version 2:

Reviewer comments:

Reviewer #2

(Remarks to the Author)

The authors have addressed all my concerns. I only have a few rather minor comments outlined below:

1. To avoid confusion please be more specific with the potential applied to the excised inside-out patches: either it is a membrane potential (-40mV), or a pipette (command) potential (40mV). The former case is usually default and typically does not require explanations, and in the latter case it should be explicitly specified.

2. Line 254:

"The MscS part of the chimera results in a higher conductance than WT-Ynal (0.1 nS4)..." Do the authors mean that the MscS part of the chimera (the pore lining helices and the C-terminus) is responsible for a higher conductance of the chimera compared to WT Ynal? If yes, I'd recommend to re-phrase this statement to avoid confusion.

3. Line 336:

"However, the flickery openings are neither inherited by MscS nor Ynal." Do the authors mean that the flickery openings are not inherent to MscS (Ynal)?

4. Line 434:

“The new findings show that pocket lipid removal cannot initiate gating in YnaI, as indicated by the fully crowded pocket, and highlight that area expansion following membrane tension is the major driving force for rearranging the protein scaffold towards a wider pore.”

I think this is a very strong statement, which is not completely supported by the data as the authors mention multiple times in the text (as well in the rebuttal major point 2) that the resolution of the pocket regions is not sufficient to unambiguously attribute electron densities to lipid molecules.

Line 594:

“Values for MscS and YnaI were taken from the literatures”. What values – tension sensitivity, unitary conductances, or both?

Dear Reviewers,

We appreciate your constructive and insightful feed-back on our manuscript and have reworked it according to your suggestions. We understand the previous, serious concerns regarding the functional data for the Ynal-MscS chimera and have added more experiments. We think that these new experiments show undoubtedly that we have measured the tension induced conductance of the chimeric channel, which reaches up to 1.2 nS, and that our new data fully support the claim that the tension required for opening follows the paddle donor and the conductivity the pore and vestibule donor.

Regarding the novelty of the structural data: We agree that several structures of Ynal are available. In particular, high-quality maps of closed Ynal exist. However, the outcome of the structure determination of mechanosensitive channels strongly depends on the reconstitution system. Here we have moved to detergent reconstitution with supplemented lipids, which allows us to resolve the outer paddle helices together with the lipids much better than in any previous reconstruction.

The open Ynal structure that we present, is unprecedented with a much wider pore and a fully resolved paddle compared to any previously published structure that was designated as open or open-like Ynal channels. The same is true for both structures of the Ynal-MscS chimera.

The interesting outcome in comparing the open and closed structures of the many different MscS-like channels is that every channel shows a somewhat different response to the reconstitution system and a different underlying conformational change, which is concomitant with pore widening. For us the structural changes that support pore-widening in Ynal and in the chimera were unexpected regarding the known data of MscS and other MscS-like channels. We believe that many more structures of the diverse group of MscS-like channels will be required to fully understand the unifying structural determinants of tension response, gating, and adaptation.

REVIEWER COMMENTS

Reviewer #1 (Remarks to the Author):

Although mechanosensitive ion channels are found in both prokaryotes and eukaryotes, they exhibit significant structural diversity. Despite recent advances in understanding their working mechanisms and the detailed dissection of several key mechanosensitive ion channels (such as MscS, K2P, OSCA, et al.), the commonalities and diversities in the features of the inductive forces of these proteins still require further elucidation through additional examples.

In the manuscript NCOMMS-24-71655 titled "Ynal exemplifies the diversity of structural gating mechanisms in mechanosensitive channels of small conductance," Vanessa J. Flegler et al. report the high resolution open and closed structures of Ynal and its chimera with MscS. Their findings show that Ynal opens through the radial relocation of the transmembrane sensor paddles, accompanied by a shortening of the pore, which contrasts with the mechanism observed in MscS. Interestingly, a thinner pore is also noted in the recently published OSCA channel (Yuanyue Shan et al doi:10.1038/s41467-024-51800-0). Additionally, the relocation of several lipids and the kink in the transmembrane region at a flexible residue are also reminiscent of the working mechanism of the OSCA channel. Also, the detergent or lipid

environment largely shapes the state of Ynal. These results seem to provide clues to the commonalities between the MscS and OSCA families.

Moreover, based on the functional data of the chimera containing both Ynal sensor paddles and the pore region from the C-terminal part of MscS, the authors propose that elements with different types of structural rearrangements can be mixed and matched within a single channel, as long as they support the common area expansion on the periplasmic side.

Overall, the results are intriguing, and the manuscript is well-written. The only shortcoming is that the existence of the open and closed structures of several MscS channel somewhat diminishes its novelty. I am happy to recommend the publication of the paper after the following comment is addressed.

We understand the shortcoming that these are not the first structures of MscS-like channels. However, as pointed out in the introduction, so far, every channel opens somewhat differently, and we will need many more structures to understand the unifying concepts of the tension response and gating in MscS-like channels.

We have addressed the comments in detail in the following.

Major/minor comments:

1. Since the importance of the flexible residues Gly149-Gly150, it is suggested to provide functional experimental evidence of their role in channel opening.

We have noticed the kinking of the pore helix TM3 at this region previously in an open-like Ynal, and data on two mutants, G149P, and G149A/G152A are already published (Flegler et al., PNAS 2020 (<https://doi.org/10.1073/pnas.2005641117>)). These experiments confirmed the importance of the ability of the pore helix TM3b to kink upon opening. Briefly, the double mutant G149A/G152A, which was designed to prevent kinking, shows a loss-of-function phenotype as its ability to protect cells in a hypoosmotic downshock assay is impaired compared to WT-Ynal. On the other hand, the mutant G149P that should stabilise the kink shows a severe growth phenotype which is in agreement with an expected gain-of-function mutation. We added the following sentence to refer to these earlier data in the discussion:

“The bending in this region has been investigated previously with two mutants: G149A/G152A, which should prevent kinking, was not able to protect cells in downshock assays, while G149P, which was designed to stabilise the kink, resulted in a gain-of-function phenotype that was reflected by a severe growth deficiency.”

2. Figure 6 and Figure S6b: There is no mention of an interaction between Leu71 and Ile124 in the previous structural analysis. It is unclear why this interaction is shown in the gating transition schematic in Figure 6. Additionally, Figure S6b shows the hydrogen bond between Lys71 and Asn8 disappearing in the open state of Ynal. This raises the question: Is residue 71 Leu or Lys?

We thank the reviewer for noticing. The residue number in Figure 6 (now Figure 7) is wrongly annotated – it is Leu41. Residue 71 is Lys71. We have corrected and replaced the Figure and corrected the figure legend accordingly.

3. SDS-PAGE in Figure S2b and Figure S4b: It is suggested to rerun the gels. The SDS-PAGE in Figure S7a mentions that the chimera expression level is low, but the bands are barely visible in the gel. The gel should be rerun and the image replaced accordingly.

We have rerun the SDS gels for WT-Ynal (condition I+ and condition III) and added the gels for the Ynal-MscS chimera (conditions I+ and condition III). This is now in Figure S8.

We purified Ynal and the Ynal-MscS chimera under condition I starting with equal amounts of cell pellets. Instead of the whole cell Western blot we have added a Western blot of the purifications of WT-Ynal and the chimera. We think this Western blot highlights that the chimera is less expressed than Ynal. However, a substantial amount of Ynal is lost during the IMAC purification giving similar amounts of purified channel at the end of the size exclusion chromatography.

4. Cryo-EM Analysis: It would be beneficial to include the detailed Cryo-EM analysis process for the closed and open Ynal-MscS chimera in the supplementary material.

We have prepared detailed cryo-EM workflows of the data processing that resulted in the closed and open Ynal, as well as in the closed and open Ynal-MscS chimera. They are now depicted in the supplement figures S10, S11, S12, and S13.

Reviewer #2 (Remarks to the Author):

Ynal exemplifies the diversity of structural gating mechanisms in mechanosensitive channels of small conductance

The manuscript by V.J. Flegler et al reports high-resolution cryo-EM structures of the bacterial mechanosensitive channel Ynal both in the closed and open states, as well as the closed and open states of Ynal-MscS chimeric channel, containing a pore-lining helix and a complete cytoplasmic cage of MscS combined with four first TM helices of Ynal. The authors highlight the unique arrangement of WT Ynal channel, which, unlike closed MscS channel, does not produce a significant membrane invagination and has a wide funnel-shaped entrance with an unusually short conductive pore when opened. Based on the closed and open structures of WT and gain-of-function A155V mutant, the authors hypothesize on a possible gating mechanism of this channel. Having obtained closed and open structures of Ynal-MscS chimeric channel, they further speculate that the pore-lining helix and C-terminus on one hand and the sensor paddle on the other have separable functions: the pore helix and C-terminus control conductance, while the sensor paddle is responsible for tension sensitivity.

Although cryo-EM structures of WT Ynal are not new, the data is of high quality, and allows to identify previously overlooked inter-molecular interactions within the oligomeric channel. However, the authors do not comment a lot on the differences between new and previously published structures (including those obtained by them earlier). The closed and open structures of Ynal-MscS chimeric channel, presented in the manuscript, illustrate an interesting, though not novel, concept of separable functions of different parts of the channel-forming protein, which is potentially applicable to many, not necessarily related, channels.

Unfortunately, in my opinion, the manuscript has a major problem, related to the functional data, because of which the conclusions presented by the authors are not justified enough. This

fact does not allow me to recommend it for publication without serious modifications. The main concerns are summarized below in points #1 and, especially, #2.

We understand that the presented functional data could not convince the reviewer that our conclusions were fully justified. We took this concern seriously and reworked large parts of the electrophysiology and the biochemistry for the chimera. In particular, we added data in a $\Delta 7$ background by using an *E. coli* strain where all seven endogenously expressed mechanosensitive channels are absent. The electrophysiological experiments proofed that the tension-induced openings are dependent on the expression of the chimera. Despite the flickery nature of the openings, we observed a conductance up to 1.2 nS in the $\Delta 7$ strain. This conductance is very close to the conductance of MscS. We conclude that the newly added functional data fully support our previous claims.

The changes are detailed below in the point-by-point reply.

1. Protein isolation and purification.

- Cryo-EM: I guess the construct included a GFP C-terminal tag, which was cleaved later, but it has not been mentioned either in the Methods or in the figure legends. In all size-exclusion chromatograms the amplitudes of the target peak P1 at ~15ml (WT Ynal on Fig.S2, Ynal-A155V on Fig.S4, Ynal-MscS chimera on Fig.S8) seem to be similar (about 1000 mAU), which indicates comparable amounts of protein.

We appreciate the reviewer's concern and now elaborated on this in more detail in the methods section. The constructs do not include a C-terminal GFP tag. Briefly, all constructs used have a C-terminal „LE-ENLYFQS-HHHHHH“ sequence, and the His₆ tag is cleaved by TEV protease after IMAC. Conclusively, the final proteins have an additional „LEENLYFQ“ sequence at the C-terminus. We cleave off the His-tag, as it reduces aggregation when concentrating Ynal for cryo-EM. This rationale is now explained in the methods section (in *Expression and purification of Ynal constructs*).

We redid small scale purifications of Ynal and the Ynal-MscS chimera starting with similar amounts of cells. We confirm that we purify similar amounts at the end of the size exclusion chromatography, although the whole cell Western blot shows a much lower expression for the chimera (see reply to next comment).

- However, the whole cell Western blot (Fig.S7a) shows very low (if any at all) amount of Ynal-MscS chimera protein (indicated by an arrow). According to the description, the same cell line (MJF641) was used for protein production for both cryo-EM and electrophysiology. The authors mention only one vector (pTrc) in the Methods section, so I suppose it was also used in the both assays. Therefore, one would expect similar amounts of protein produced and, therefore, similar intensities of the bands. This obviously is not the case. In fact, the quality of the gel seems to me insufficient to tell if there is a real band for the chimera protein on Fig.S7a. The authors did not mention the antibodies they used either (that were probably anti-6His). Also, why the chimera lane is separated from the rest ($\Delta 7$, MscS, Ynal, and the ladder) of the lanes? Was it obtained separately, under different conditions, on a different day, etc and then combined with an image of another gel? The authors should comment on this.

We thank the reviewer for noticing. Indeed, the low expression level for the Ynal-MscS chimera, that is observed in the whole cell Western blot, contradicts the high protein yield after purification. We investigated this discrepancy and did a Western blot of the purifications of Ynal and the chimera with following samples applied: equal amounts of the whole cell pellets used for purification resuspended in PBS buffer, and flowthrough, wash fraction and E1 elution fraction of the IMAC. Equal volumes of all samples were applied. In the Western blot, the signals of the E1 elution fractions of Ynal and the chimera are comparably intense, but there are no signals visible for the chimera's cell pellet, flow-through and wash fraction, but for Ynal. Considering that only 10 μ l were applied on the gel of the flow-through and wash fractions out of 20 ml flowthrough in total and 35 ml wash fraction in total, we conclude that a substantial amount of Ynal is lost during IMAC. The chimera on the other hand, is worse expressed, but more efficiently purified.

As this Western blot is more informative than the whole cell Western blot, we included this blot in Fig. S8 (Purification of the closed and open Ynal-MscS chimera).

The previously included whole cell Western blot was also originally one experiment and one SDS-PAGE (see below), but this Western blot included also various other chimeric constructs, which were not further pursued for this paper and therefore excluded from the manuscript for reasons of clarity. We removed this whole cell Western blot from the manuscript, because the low expression level of the chimera is also obvious in the whole cell pellet sample of the standard Western blot that we have added in Figure S8d.

2. The concern about the presence of Ynal-MscS chimeric channel in the membranes of *E. coli* spheroplasts used for patch-clamping, raised by the reasons listed in the previous point, is very serious also because of the following:

- The authors state, that (lines 291-292), that “the chimera provides protection in hypoosmotic shock assay, despite being only weakly overexpressed (figure s7a,b)”. The same statement is also re-iterated in Figure 5a (“Protection in downshock” in the table). However, neither in Figure 7, nor anywhere else in the manuscript there are data or references supporting this statement. This also applies to MscS and Ynal (Figure 5a, table).

We have included the downshock experiments for the Ynal-MscS chimera together with WT-Ynal, WT-MscS and the MJF641 ($\Delta 7$) strain to Figure S7 and extended the methods section accordingly. Triplicates of the downshock assays were performed, showing that expression of the Ynal-MscS chimera protects *E. coli* $\Delta 7$ cells upon a hypoosmotic downshock.

- The same table indicates unitary conductance of Ynal-MscS chimera of ~ 1 nS, which is about 3 times lower than unitary conductance of MscL under the same conditions (about 3.3 nS), however Figure 5b demonstrates “chimera” unitary currents of about 1/10 (or even less) of those of MscL, which clearly contradicts the data presented in the table.
- This also applies to Figure S7b and its legend. The channel openings labeled as Ynal-MscS (“#”) definitely have the conductance much lower than 1 nS, because the openings of WT MscL (“*”) should be about 3 nS.
- The reasons why the authors attribute two obviously different types of openings (very flickery and unstable in the upper and lower panels of Figure 5b and more stable table-like opening in lower panel of the same figure) are also unclear. It is not uncommon to see spontaneous low amplitude flickery activity when membrane is subject to near-lytic tensions, activating MscL (which is also present in the figure).
- Figure 5b and Figure S7b legends indicate the patch-clamp experiments were done in MJF429 (Δ MscK, Δ MscS) cell line, which still expresses endogenous MscL, Ynal itself (!), and three more (!) endogenous mechanosensitive channels of *E. coli*. Taking into account this background and the absence of ~ 1 nS conductances presented in Figures 5b/S7b, it is more than likely that the electric activities, presented in these figures are not caused by Ynal-MscS chimera.

We agree with the reviewer that the activities in MJF429 could be misleadingly assigned to the Ynal-MscS chimera although the chromosomal expression of Ynal, YbiO and YjeP is low, and activities are rarely seen. We repeated and revised the patch-clamp experiments:

New electrophysiology experiments in MJF641 (“ $\Delta 7$ ” – all 7 chromosomal mechanosensitive channels from *E. coli* deleted) show tension dependent activities, We also conducted the patch-clamp experiments on MJF641 without expressing the chimera. This shows no activity and confirms that the activities belong to the Ynal-MscS-chimera. The new experiments are shown in Figure 5c.

The experiments on MJF429 show that the pressure required to open the Ynal-MscS chimera is close to that of MscL.

To address the concerns regarding the determination of the conductance of the Ynal-MscS chimera we analysed the activities of the chimera in MJF641. Here all activities can be attributed with certainty to the chimera. All openings show very short open dwell times. Although it is obvious that this channel can open with a significantly larger conductance than WT-Ynal, it is not possible to give a unitary conductance because of the short dwell times. We do not fully resolve the opening in time and therefore the measured conductance are lower estimates of the unitary conductance. Upon very high pressure (Figure S7a), the current trace plateaus at ~ 50 pA which might reflect the achievable full conductance of the Ynal-MscS chimera.

- Channel adaptation (Figure 5a table) is not illustrated/supported by experimental data; if taken from the literature – no references are provided.

We have not measured adaptation for the Ynal-MscS chimera, because of its short dwell times. We removed the row from the table in Fig. 5b.

All references for the table in Figure 5b are now given in the figure legend of Figure 5:

MscS conductance:

Levina N, Töttemeyer S, Stokes NR, Louis P, Jones MA & Booth IR (1999) Protection of *Escherichia coli* cells against extreme turgor by activation of MscS and MscL mechanosensitive channels: Identification of genes required for MscS activity. *EMBO J* 18: 1730–1737

MscS gating threshold:

Edwards MD, Li Y, Kim S, Miller S, Bartlett W, Black S, Dennison S, Iscla I, Blount P, Bowie JU, *et al* (2005) Pivotal role of the glycine-rich TM3 helix in gating the MscS mechanosensitive channel. *Nat Struct Mol Biol* 12: 113–119

MscS observed pore lipids:

Reddy B, Bavi N, Lu A, Park Y & Perozo E (2019) Molecular basis of force-from-lipids gating in the mechanosensitive channel mscS. *Elife* 8

Zhang Y, Daday C, Gu R-X, Cox CD, Martinac B, de Groot BL & Walz T (2021) Visualization of the mechanosensitive ion channel MscS under membrane tension. *Nature*: 1–6

Flegler VJ, Rasmussen A, Borbil K, Boten L, Chen H-A, Deinlein H, Halang J, Hellmanzik K, Löffler J, Schmidt V, *et al* (2021) Mechanosensitive channel gating by delipidation. *Proc Natl Acad Sci U S A* 118: e2107095118

Ynal conductance and pressure threshold:

Edwards MD, Black S, Rasmussen T, Rasmussen A, Stokes NR, Stephen T-L, Miller S & Booth IR (2012) Characterization of three novel mechanosensitive channel activities in *Escherichia coli*. *Channels* 6: 272–281

Summarizing the arguments presented in points #1 and #2: all the functional data on Ynal-MscS chimera, presented in the manuscript seems to be mostly irrelevant and therefore can't be used for justification of the proposed Ynal gating mechanism (as stated, for example, in the two last sentences of the Abstract, lines 26-32). Hence, a substantial part of the manuscript's conclusions remain unsupported (by experimental data) and represent rather speculations than established facts.

With the new experiments outlined above, we could add substantial, experimental support for our claims. Most importantly we showed that the activities stem from the Ynal-MscS-chimera by demonstrating activity in the “ $\Delta 7$ ” background and are therefore no longer irrelevant.

Minor:

1. Lines 42-43: “...but release only small solutes with a very small flux”. That is not completely true, as, for example, WT MscS, which is naturally expressed at quite density, has a unitary conductance of ~ 1.2 nS, which is only 3 times lower than the conductance of MscL. Therefore, the fluxes mediated by MscS are rather big.

We rephrased this sentence in the introduction.

2. Lines 49-51: would be nice to have at least a reference here.

We added the following three references in the introduction:

Edwards MD, Black S, Rasmussen T, Rasmussen A, Stokes NR, Stephen T-L, Miller S & Booth IR (2012) Characterization of three novel mechanosensitive channel activities in *Escherichia coli*. *Channels* 6: 272–281

Flegler VJ, Rasmussen A, Rao S, Wu N, Zenobi R, Sansom MSP, Hedrich R, Rasmussen T & Böttcher B (2020) The MscS-like channel Ynal has a gating mechanism based on flexible pore helices. *Proc Natl Acad Sci U S A* 117: 28754–28762

Hu W, Wang Z & Zheng H (2021) Mechanosensitive channel Ynal has lipid-bound extended sensor paddles. *Commun Biol* 4: 602

3. Lines 79-80. This point of view is quite outdated. For example, the recent publications of structures of MscS-related plant mechanosensitive channels FLYC1 and MSL10 indicate the lack of substantial membrane deformations and rather mild conformational changes during gating of these channels (Jojoa-Cluz et al, 2022; Zhang et al, 2023). Obviously, they very unlikely to share the same gating mechanism with bacterial MscS.

We refrained from explicitly mentioning the gating mechanisms of these two plant MscS-like channels due to the lack of structure-guided gating transitions. For FLYC1, the authors suggested that their single obtained conformation is a near-closed or subconducting state. Wildtype-MSL10, on the other hand, was obtained only in an open conformation, and a non-conducting state was observed by a pore point mutation, which induces a side chain reorientation that leads to narrower hydrophobic gate in the pore but does not show any further rearrangements. Hence, it is still elusive whether this obtained non-conducting state also resembles the true closed conformation of WT-MSL10.

Yet, we agree that the plant MscS-like channels FLYC1 and MSL10 do not display local membrane deformation, and are indeed structurally diverse from bacterial MscS, especially with respect to their cytoplasmic linker domains which can adopt up and down conformations. Thus, they will likely not share the gating mechanism of MscS. Therefore, we followed the reviewer’s concern and removed the statement.

4. Lines 86-87. As mentioned above, this statement is not confirmed by experimental data.

Our revised and extensive electrophysiological experiments show that the MscS donor pore in the Ynal-MscS chimera supports a conductance, which is significantly higher than the conductance of WT-Ynal. We accept that the short dwell time of the openings of the chimera is a feature that is not inherited of WT-MscS, so we rephrased this statement at the end of the introduction and focus it now on the structural observations.

5. WT Ynal structures, although are of high resolution and quality, are not new. The authors unfortunately do not discuss the advantages of and new details provided by these newly obtained structures.

The new high-resolution structure of closed Ynal enabled more detailed insights into the intramolecular hydrogen-bonding network of the sensor paddles, which is crucial to understand why the sensor paddles move as a rigid body, also in the Ynal-MscS chimera. We have added a sentence to highlight this fact. Furthermore, the map resolved a multitude of pocket ligands better than previous maps, which allowed to follow their reorganisation upon opening. Both statements are addressed in the manuscript.

We regard the structure of open Ynal as new. Earlier, we had published a structure that we dubbed “open-like” (Flegler et al. PNAS 2020). This open-like structure was very incomplete, low resolution and did not resolve the paddles. As we now know, the paddles were distorted by the interactions with the harsh environment of the surrounding polymer. However, the open-like structure already indicated the kinking at the GGIGG motif. We reference this in our manuscript.

The open structure that we present here shows a different conformation with a much wider pore than the open-like channel (Flegler et al. 2020). It also resolves the paddle helices in the less disruptive environment of the detergent micelle. This allowed us to generate the first (almost) complete model of the open Ynal,

6. Figure 1b. The resolution map, presented as an overlay of pseudo-colored density map and the protein structure is a bit difficult to comprehend. I'd suggest to take the usual approach and to demonstrate the pseudo-colored 3D density map of the whole channel instead.

We followed the reviewer's suggestion and exchanged Figure 1b.

7. The authors do not provide any figures illustrating the water-accessible pore profiles (obtained with HOLE or similar software) for any of the structures. It makes it difficult to appreciate the structural rearrangements, resulting in the channel opening for example. Also (as in line 175: “...increase in diameter from ~9Å to 22Å in open Ynal...”) the references to the structural rearrangement are quite obscure as they are neither illustrated in figures, nor they specify the exact residues where the measurements were taken. This also applies to several other statements, such as, for example, found in lines 223-224: “...the effective pore length is reduced from ~35 Å to ~20 Å.” There is no panel in Figure 3 illustrating this, there is also no direct indication between which residues this distance was measured.

We applied HOLE (Smart et al., J Mol Graph 1996) implemented in Coot 1.1.13 to determine the radius profiles of our open and closed Ynal pores and closed and open chimera pores. For this,

we chose the heights of the residues G139 at the periplasmic side of the pore and D162 (for Ynal) or G162, respectively (for the chimera) at the cytosolic side of the pore. We included the radius profiles in Figure 3a and Figure 6a, respectively. Furthermore, we added a scale bar (from ChimeraX) to Figure 3a.

8. Line 286: pressure ratio of 1.04 seems to be mistakenly labeled as “MscS”. Also, the Methods only describe how the pressure ratio of MscL:chimera was obtained. What is the origin of the pressure ratios of MscL:Ynal and MscL:MscS? Were they measured by the authors or taken from the literature? In the latter case, the references should be presented.

We fixed the given pressure ratio.

We added the references for the pressure ratios MscL:Ynal and MscL:MscS:

Gating threshold of Ynal:

Edwards MD, Black S, Rasmussen T, Rasmussen A, Stokes NR, Stephen T-L, Miller S & Booth IR (2012) Characterization of three novel mechanosensitive channel activities in *Escherichia coli*. *Channels* 6: 272–281

Gating threshold of MscS:

Edwards MD, Li Y, Kim S, Miller S, Bartlett W, Black S, Dennison S, Iscla I, Blount P, Bowie JU, *et al* (2005) Pivotal role of the glycine-rich TM3 helix in gating the MscS mechanosensitive channel. *Nat Struct Mol Biol* 12: 113–119

9. There are a few sentences in the manuscript that look to me like containing duplicate, wrong, or missing words (lines 142, 178, 253-254, 558).

We fixed this.

Dear Reviewer #2,

Thank you for your helpful remarks and suggestions. We acknowledge that we needed to change our title to a more appropriate one. The new title now reflects the Ynal-MscS chimera as well and we extended the corresponding sections in the results and discussion section. We have included the requested analyses and figures and elucidated sentences and paragraphs that have not been entirely clear.

In the following point-by-point reply, our line references refer to the tracked-changes document.

Reviewer #2 (Remarks to the Author):

Major:

1. Reading the manuscript, I am still struggling to find a comprehensive discussion of the point outlined in the title, namely, “the diversity of gating mechanisms in mechanosensitive channels of small conductance”. I would suggest the authors either to change the title to reflect the main points of the manuscript or to provide a more detailed review of MscS-like channels gating mechanisms.

We changed the title to “**Mechanosensitive Channel Engineering: A Study on the Mixing and Matching of Ynal and MscS Sensor Paddles and Pores**”. This title fits more to the insights we gained from our structural and functional analyses on Ynal and the Ynal-MscS chimera.

2. In the Ynal closed state the authors identify 8 alkyl chains per subunit (lines 113-120). In the same time, in the open state only six alkyl chains per pocket (line 218) are identified, which suggests that one lipid molecule leaves the pocket upon/during channel opening. This observation contradicts the authors speculation that the pocket lipids cannot leave the pocket (or exchange with the lipid bulk of the membrane), as stated in lines 407-409. On the other hand, the authors state, that a) “the pockets in the cytosol were continuous with the grooves between the paddles implying that lipids can enter the pockets from the reservoir of the membrane” (lines 127-129) and b) the (membrane) “ lipids might push against the bridge and become continuous with the pocket lipids” (lines 396-399), which is very confusing. I would suggest the authors to review the gating model description to make it more comprehensive.

We understand the reviewer’s concern and try to clarify this point in the following. The addition of lipids to the purification buffers proved beneficial for the map quality of the transmembrane part of closed Ynal. However, for the purification of the open conformation, we omitted the additional lipids in the buffers and needed to increase the DDM concentrations. Therefore, it is not surprising that the map quality of the transmembrane part of open Ynal is below that of closed Ynal, which explains why also less alkyl chains are resolved. We emphasize that not- or not well resolved features might be present, nonetheless.

The statement “the pockets in the cytosol were continuous with the grooves between the paddles implying that lipids can enter the pockets from the reservoir of the membrane” (lines 129-131) refers to the closed conformation, which does not have the hydrophobic bridge (Leu71-Ile124) like the open conformation. Thus, the grooves between the paddles of closed Ynal are passable for lipids. In the open state, the hydrophobic bridge is formed, and the remaining lipids of the pockets can no longer equilibrate with lipids of the membrane.

We further followed the reviewer's suggestion and rephrased the gating model description for clarification.

3. In the Results section the authors do not comment on the localization of the lipid pockets in YnaI-MscS chimera (closed or open), neither they mention what lipid-like densities were identified in the pockets. This is very strange as the role of lipids in MscS and MscS-like channels gating is one of the main points of the manuscript.

We thank the reviewer for noticing. Indeed, we had elaborated on the poor lipid coordination only in the discussion section. Now we have included a description in the results section (lines 337-342) and also extended the figure S9 (c and g).

Electrophysiology of YnaI-MscS chimera:

1. Open and closed conformations of a YnaI-MscS chimera. I'd suggest to start this part with the experiments done in MJF641 cell line and then to move to MJF429, as the previously unknown (YnaI-MscS chimera) unitary channel properties are characterized on the clean background of MJF641 first. Then, based on these results, it is possible to discriminate YnaI-MscS from the rest of potential endogenous MscS-like activities in MJF429.

Also, please provide the rationale why the membrane stretch-activated currents in MJF429 may not (or may) be caused by endogenous mechanosensitive channels, like, for example, YbiO, having similar to MscS unitary conductance, and lower tension sensitivity (doi: 10.4161/chan.20998).

I would also recommend to present the all-point histogram for figure s7 (and maybe figure 5c) raw trace data to illustrate the most probable conductance level(s) of YnaI-MscS chimera.

We followed the reviewer's suggestion and rearranged the electrophysiology part of the YnaI-MscS chimera (lines 253-289). Comprehensively, the experiments on the MJF641 ("Δ7") strain that overexpresses the chimera, together with the reaffirmation that no activities are seen in the Δ7 strain alone, revealed a current of about 50 pA (corresponding to 1.3 nS at +40 mV) for single channel openings. We thank the reviewer for recommending the all-point amplitude histograms of the traces recorded of the Δ7 strain overexpressing the chimera, which confirmed our observation that the most probable current of a single channel opening is around 50 pA. The exactly determined values were 50.1 ± 1.7 pA (maximum value \pm standard error) and 26.4 ± 4.1 pA (FWHM \pm standard error) for the trace shown in figure 5c, and 50.5 ± 0.6 pA (maximum value \pm standard error) and 22.3 ± 1.6 pA (FWHM \pm standard error) for the trace shown in figure S7a. The all-point histograms are depicted next to their corresponding traces in figure 5c and in figure S7a.

To determine the pressure threshold to MscL, we used the strain MJF429, which only has the genes of MscS and MscK eliminated. Accordingly, the MscS-like channels YnaI, YbdG, YbiO and YjeP are still endogenously expressed. The reviewer's initial concern how to discriminate the YnaI-MscS chimera from the channels was valid, and we have included the rationale why the observed flickering currents that occur at a similar pressure required to open MscL cannot come from these endogenously expressed channels. First, as already mentioned, the chimera's most probable conductance 1.3 nS, and YnaI, and YjeP have a much lower conductance (0.1 and 0.3 nS). YbdG has no detectable conductance. YbiO conducts 1 nS but shows a stable staircase-like trace and needs less pressure to open. Furthermore, YbiO is barely expressed

and/or assembled in the membrane. This reasoning together with necessary references are now provided in lines 270-289.

Minor:

Line 141: "...WT-Ynal and Ynal155 were indistinguishable..." On figure s4d, probably due to the graphical artifact (depth occlusion?) blue TM1 does not seem to be overlaid with a green one; maybe RMSD value will be more suitable here.

We exchanged figure S4d, and now we show only one pair of chains aligned for clarity. We omitted the depth cueing. The RMSD of the two models is 0.92 Å, and the major differences occur in weaker resolved loops and the outermost paddle helix TM(-2).

Lines 146-148: "the pockets in the open state are pulled into the plane of the cytosolic membrane leaflet and the pocket lipids align with the lipids of the cytosolic leaflet". I'd suggest to illustrate this conformational change with a figure (and to estimate the displacement in Å) as this feature is important for the further discussion of Ynal gating.

We included a figure S6g to illustrate the displacement of the pockets with respect to the cytosolic membrane leaflet. It shows that the distance of the pockets to the membrane is reduced by approx. 5 Å in the open state. As the pockets are not pulled completely in the membrane, we changed the sentence to "the pocket lipids in the open state are pulled towards the plane of the cytosolic leaflet..." (line 149). The alignment with the lipids of the cytosolic leaflet refers to the change of the orientation of the pocket lipids upon opening. This is already illustrated in figure 4.

Figure 3: the pore profiles plot is erroneously labeled as figure 3b.

We corrected this error – it is figure 3a.

Figure 4: please label the alkyl chains with a brighter color, as they are really difficult to see. Line 213: "figure 3f" – there is no such figure in the manuscript.

We changed the colour of the alkyl chains to a brighter one and exchanged the figure 4.

Figure 3f" was changed to figure 3b.

Lines 215-217: "most inter-subunit interactions in the TM region of Ynal are lost" – I guess, the authors mean the interactions between the paddles of the subunits? I guess the TM3-TM3 interactions should be preserved, otherwise the pore would become unstable.

We also refer to the inter-subunit interactions in the pore, but explicitly the residues highlighted in figure 3b. Therefore, we now changed the wording ("many" instead of "most") (line 218). But figure 3b clearly shows that many prominent interactions of closed Ynal are lost in the open conformation. Pore stability is conferred by hydrophobic packing between adjacent TM3 helices.

Lines 303-304: “However, the flickery openings are neither inherited by MscS nor Ynal” and are probably due to what? Maybe some speculations/hypothesizing on the origin of such a behavior?

A hypothesis on the origin of the flickering chimera openings was already presented in the discussion section (lines 397-408) and is now also briefly addressed in the results section (lines 337-342). Briefly, we speculate that the lipid coordination is disturbed in the chimera, resulting in the loss of the ability to open stably. The chimera’s hydrophobic pockets derive from both donor channels, and it is conceivable that this arrangement does not favour stable lipid coordination. This hypothesis agrees with our cryo-EM maps where we observed that the pocket ligand densities are not, or much worse resolved, respectively.

Following the studies of Park *et al.* (2023), without the energetic contribution from aligning the pocket lipids in the open state with the bilayer the open conformation is less stable. This is accompanied by quick closings, observable as flickering activities.

Figure 6a: I’d suggest to add WT MscS (open and closed conformations) pore profile to the plot alongside with the Ynal-MscS chimera’s. Could the differences give a hint of the reasons why the chimera activity is different from WT MscS?

While the pores of the closed conformations of MscS and the chimera are almost identical, the open chimera pore is funnel-shaped and thus clearly different from the open MscS pore. Yet, we feel our initial hypothesis is more likely – that the novel pocket composition results in impaired lipid coordination which in turn abolishes the chimera’s ability to sustain open (see above). Therefore, we refrained from showing the pore profile of MscS along with the chimera’s.

Lines 360-363: Maybe I am missing something, but I do not see any description or figures illustrating hydrophobic pockets (and lipid-like densities in them) of the Ynal-MscS in Results.

The description and reference to figure S9 c and g is now added (lines 319, 323, 338). We further extended figure S9c and g to better illustrate the weak pocket lipid densities.

Lines 379-380: “reminiscent of K⁺ channels, which only have a relevant pore length of 12Å”. I do not think it is a valid comparison. The authors refer to the length of highly conserved selectivity filter, however the pores of potassium channels are far more complex and demonstrate constrictions, that in many cases are almost as narrow as their selectivity filters, of which Kir2.X channels (<https://doi.org/10.1126/sciadv.abq8489>) are a good example. Although the authors’ reference might be true for some potassium channels, in general their pores cannot be approximated by a simple funnel shape, similar to that the authors suggest for Ynal. Also, as noted by the authors, Ynal ion selectivity (or rather ion preference), unlike potassium channels, is dictated by the side portals of its cytoplasmic cage, but not by the constrictions of the transmembrane pore.

We thank the reviewer for this clarification and agree that the comparison of selectivity filter of the K⁺ channels with Ynal is unsuitable. Therefore, we have removed this comparison and the references.

Lines 385-386: “...cannot reduce the kink dividing the helices TM3a and TM3b and cannot bend the helix TM3a outwards”. What is the force preventing this conformational change? The authors

postulate independent functions of the pore and the paddle domains, so the MscS-derived pore of the chimeric channel is probably expected to adopt “open-MscS” conformation?

The chimera inherited the pore of MscS and therefore misses the di-glycine hinge of the YnaI pore, which is crucial for bending the helix TM3a in this region. Although the open YnaI-MscS chimera shows a novel pore, it exhibits more features of the open MscS conformation than the open YnaI conformation regarding not only the kink at Gly149-Gly150. However, the origin of the ability to reduce the kink dividing the helices TM3a and b, resulting in one continuous straightened helix, remains unknown.

Miscellaneous:

Line 22: Despite several structures...We changed the wording.

Line 93: ...and it(?) provides... “it” is introduced.

Line 157: ...which interact with each other(?) via Van der Waals... Done.

Line 367: unstable Done.

Line 388: which gives rise Done.

Point-by-point answers to Reviewer #2:

The authors have addressed all my concerns. I only have a few rather minor comments outlined below:

1. To avoid confusion please be more specific with the potential applied to the excised inside-out patches: either it is a membrane potential (-40mV), or a pipette (command) potential (40mV). The former case is usually default and typically does not require explanations, and in the latter case it should be explicitly specified.

We refer to the pipette potential. It is given correctly.

2. Line 254:

“The MscS part of the chimera results in a higher conductance than WT-YnaI (0.1 nS4)...” Do the authors mean that the MscS part of the chimera (the pore lining helices and the C-terminus) is responsible for a higher conductance of the chimera compared to WT YnaI? If yes, I'd recommend to re-phrase this statement to avoid confusion.

We agree that this statement was not phrased clearly. We changed it to “Due to the MscS part, the chimera exhibits a higher conductance than WT-YnaI (0.1 nS4)”.

3. Line 336:

“However, the flickery openings are neither inherited by MscS nor YnaI.” Do the authors mean that the flickery openings are not inherent to MscS (YnaI)?

Exactly. Both wt-MscS and wt-YnaI open stably. We added a side sentence: “However, the flickery openings are neither inherited by MscS nor YnaI, as these channels open stably.”

4. Line 434:

“The new findings show that pocket lipid removal cannot initiate gating in YnaI, as indicated by the fully crowded pocket, and highlight that area expansion following membrane tension is the major driving force for rearranging the protein scaffold towards a wider pore.”

I think this is a very strong statement, which is not completely supported by the data as the authors mention multiple times in the text (as well in the rebuttal major point 2) that the resolution of the pocket regions is not sufficient to unambiguously attribute electron densities to lipid molecules.

The cryo-EM maps of both closed and open YnaI revealed fully crowded pockets, although the headgroups could not clearly be assigned for many. For reasons of clarity, we decided to only indicate the hydrophobic tails by C₁₂ alkyl chains, which are well resolved in the maps. The clear tail densities show that the pockets cannot be empty. This rationale is evident from the text.

Line 594:

“Values for MscS and YnaI were taken from the literatures”. What values – tension sensitivity, unitary conductances, or both?

We refer now explicitly to the conductance values and gating thresholds: “Values (conductance, gating threshold) for MscS and YnaI were taken from the literatures^{4,7,50}.”